# From Corpora to Causality: Unveiling Causal Comprehension in Large Language Models

## Abstract

This study investigates the efficacy of Large Language Models (LLMs) in causal discovery. Using newly available open-source LLMs, OLMO and BLOOM, which provide access to their pre-training corpora, we explore three research questions aimed at understanding how LLMs process causal discovery. These questions focus on the impact of memorization versus generalization, the influence of incorrect causal relations in pre-training data, and the role of contexts of causal relations. Our findings indicate that while LLMs are effective in recognizing causal relations that occur frequently in pre-training data, their ability to generalize to new or rare causal relations is limited. Moreover, the presence of incorrect causal relations significantly undermines the confidence of LLMs in corresponding correct causal relations, and the context of a causal relation markedly affects the performance of LLMs to identify causal relations. This study shows that LLMs possess a limited capacity to generalize novel causal relations. It also highlights the importance of managing incorrect causal relations in pre-training data and integrating contextual information to optimize LLM performance in causal discovery tasks. [1]

## 1 Introduction

Identification and understanding of causal relations hold fundamental importance in human cognition and science, as those relations form the basis of causal models, which are utilized to answer observational, interventional, and counterfactual questions (Zanga et al., 2022; Wan et al., 2024). The task of identifying causal relations among a set of random variables is known as *causal discovery*, where a random variable may refer to an event in daily life, a medical treatment, or a drug effect, etc. (Pearl, 2009; Peters et al., 2017; Nogueira et al., 2021). For decades, various statistical methods have been developed to identify causal relations from observational or interventional data Heckerman et al. (1995); Chickering (2002); Koivisto & Sood (2004); Mooij et al. (2016a). However, algorithms that can accurately recover true causal structures from observational data remain elusive. Neal (2020).

With the rise of Large Language Models (LLMs), recent studies exploit the potential of LLMs for causal discovery by evaluating them on benchmark datasets Willig et al. (2022); Ban et al. (2023). Closed-source LLMs, such as GPT-3 and GPT-4, surpass the state-of-the-art (SOTA) statistical methods on several publicly available datasets (Kıcıman et al., 2023). However, Romanou et al. (2023) notice both GPT-3 and GPT-4 have a performance drop on the causal relations involving real-world events occurring post-Jan 2022, compared to the ones before Jan 2022. Kıcıman et al. (2023) find out that given part of a data table in the Tübingen cause-effect pairs dataset (Mooij et al., 2016b), GPT-4 can recover 61% of the remaining part. Zečević et al. (2023) conjecture that *LLMs may just recall causal knowledge in their large pre-training corpora by acting as "causal parrots"*. However, there are no solid experiments to verify to what extent *memorization* and *generalization* affect model performance in causal discovery tasks because the pre-training corpora of those LLMs are not accessible and the high-performing LLMs are closed-source.

The recently released open-source LLMs OLMO and BLOOM make their respective pre-training corpora Dolma and ROOTS publicly available Groeneveld et al. (2024); Workshop et al. (2023).

---

[1] The code and data are available at `https://anonymous.4open.science/r/causality_llm-5FD3`

This provides the opportunity for us to investigate the correlations between model outputs and the frequency of relations mentioned in their pre-training corpora. In this work, we focus on three research questions and try to conduct experiments to answer them. *RQ 1) What is the difference in performance on causal discovery tasks for LLMs when recognizing relations through memorization compared to inferring them through generalization? RQ 2 How does the occurrence of incorrect causal relations affect LLMs performance in causal discovery?* and *RQ 3 How does the context of a causal relation influence LLM performance in causal discovery tasks?*

Our experiments reveal the following findings.

- Although LLMs are proficient at recognizing causal relations through memorization, their ability to generalize novel causal relations is highly limited. This limitation poses significant challenges for deploying LLM-based causal discovery methods in scenarios where causal relations are rarely or not included in their pre-training data.

- The presence of incorrect causal relations, such as the reversal of correct causal relations, adversely impacts LLMs' confidence in identifying correct causal relations. This finding highlights the necessity of minimizing conflicting causal information in pre-training datasets to enhance the performance of LLMs.

- The validity and strength of causal relations can vary significantly across different contexts. This variability suggests that LLM-based causal discovery methods should incorporate the context of causal relations as input to ensure accuracy, particularly to avoid misleading contexts that could substantially degrade performance.

## 2 BACKGROUND

Causal discovery aims to identify causal relations among a given set of random variables. For each pair of variables $X$ and $Y$, it identifies whether $X \leftarrow Y$, $Y \leftarrow X$, or there is no causal influence between them, where $\leftarrow$ denotes the direction of causality. The traditional algorithms for this task are statistical methods that perform causal discovery on tabular data, which are capable of unveiling previously unknown or uncertain causal relations that are not *explicitly* mentioned anywhere in text (*e.g.,* "sea level pressure causally influences zonal wind at 10 m" Huang et al. (2021)). In contrast, prior NLP methods focus on either extracting mentions of known causal relations from documents Yang et al. (2022) or answering questions related to causality Oh et al. (2013). The gold standard for causal discovery is experimental approaches such as randomized controlled trials and A/B testing Fisher (1935). However, such experiments are often not feasible due to ethical or financial constraints, which necessitates the use of alternative methods that rely solely on statistics collected from observational data.

The statistical causal discovery methods are conventionally categorized into constraint-based methods, such as Peter and Clark (PC) Spirtes et al. (2000) and inductive causation (IC) Pearl (2009), and score-based methods Heckerman et al. (1995); Chickering (2002); Koivisto & Sood (2004); Mooij et al. (2016a). Those methods rely on statistics calculated from tabular data to infer causal graphs, in which random variables are depicted as nodes and their causal relations are represented as edges. However, a significant drawback of these approaches is their dependency on extensive data collection to construct reliable tabular data, a process that can be both time-consuming and costly. Furthermore, a theoretical limitation of these statistical methods is their inability to precisely predict ground-truth causal graphs, unless strong assumptions are made. Instead, they typically yield an equivalence class of true causal graphs Spirtes et al. (2000); Pearl (2009).

Recent advances of LLMs provide new opportunities to tackle the task without accessing tabular data by formulating it as a pairwise causal relation prediction task Kıcıman et al. (2023); Zečević et al. (2023); Long et al. (2022). Given a pair of variable names, an LLM is instructed to identify which is the cause and which is the effect using prompts Kıcıman et al. (2023); Zečević et al. (2023), by distilling such knowledge directly from the LLM. However, the reliability of such methods is under scrutiny. Zečević et al. (2023) argue that LLMs are *"causal parrots"*, which may depend on *memorization* to recall the causal relations present in their training data. In other words, LLMs may not *generalize* well to detect causal relations that seldom or never occur in pre-training data. If this argument holds, LLMs may primarily excel at reproducing causal relations known in their training data rather than uncovering novel ones. However, there is no solid empirical justification

of this argument because prior works employ either commercial LLMs or open-source LLMs that have no access to their training data. The current techniques for understanding and investigating memorization in LLMs are still in their infancy Speicher et al. (2024).

# 3 METHODOLOGY

We aim to investigate the key limitations of LLMs for causal discovery by answering three research questions. The first research question aims to collect stronger empirical evidence to verify the "causal parrots" hypothesis. The second research question investigates to what extend the presence of incorrect causal relations in the training data, which are oriented in the opposite direction to their true counterparts, influence the performance of LLMs. Instead of only feeding two variable names to LLMs, the last research question is concerned with the *first* quantitative study on how the context of a causal relation impacts the predictive performance of LLMs.

Unlike prior works, we collect evidence of memorization from the pre-training datasets of LLMs and investigate their statistical properties in relation to LLMs' predictive performance. As it is almost infeasible to collect all mentions of a causal relation from a dataset, we curate a synthetic causal relation dataset to further investigate to what extent LLMs can generate to unseen causal relations. Herein, we select OLMo-7b-Instruct and BLOOM-7b1, which are the LLMs that have their pre-training data publicly available Groeneveld et al. (2024); Workshop et al. (2023).To determine whether LLMs primarily rely on memorization or generalization, we classify causal relations into various occurrence intervals, ensuring that each interval contains a comparable number of relations. We then assess the LLMs' performance in recognizing causal relations across these defined intervals. Evaluations are performed by transforming causal relations into yes-no questions, such as "does smoking cause lung cancer?". We employ accuracy and F1 score metrics to assess performance. If LLMs mainly utilize memorization to identify causal relations, we anticipate observing high accuracy and F1 scores for relations that frequently occur in the pre-training data, with a notable decline in performance for less frequently occurring relations. This experimental method aligns with the approaches stated in Razeghi et al. (2022).

In addition to examining the frequency of causal relations, we also investigate how the presence of incorrect causal relations impacts LLMs' confidence in corresponding correct causal relations. For example, we want to explore how the occurrence of "lung cancer causes smoking" might affect an LLM's confidence in the correct relation "smoking causes lung cancer." To this end, we have devised a novel experimental setup. We assess the confidence of LLMs in correct causal relations under varying frequencies of corresponding incorrect causal relations. We hypothesize that a higher presence of incorrect causal relations diminishes the LLMs' confidence in the correct causal relations. The confidence level of the LLMs is measured by the proportion of responses that affirm the correct causal relation out of multiple generated responses for one query.

Due to the impracticality of exhaustively retrieving all semantically equivalent mentions of target causal relations in pre-training data, we create new pre-training corpora including synthetic causal and incorrect causal relations. These relations, such as "blaonge causes goloneke," utilize terms that do not exist in the original pre-training corpora. We then integrate these synthetic relations into the pre-training data at various frequencies. This approach allows us to re-evaluate our experimental results on real-world causal relations, thereby validating the reliability of our findings under controlled conditions.

One distinction between causal discovery and numerical reasoning tasks Razeghi et al. (2022) is context dependency. Numerical reasoning, such as $3 + 4 = 7$, exhibits consistency across various contexts. However, causal relations might not have this consistency. For example, the causal relation "rain causes flooding" may be true during a heavy downpour in a city with poor drainage but may not be true during light rain in areas with good drainage systems. Therefore, we assess the performance of LLMs on causal discovery across varying contexts. For each selected causal relation from human-annotated datasets, we employ GPT-4o to generate five positive contexts that affirm the relation and five negative contexts that challenge it. The LLMs' ability to recognize these causal relations is then evaluated in these contexts.

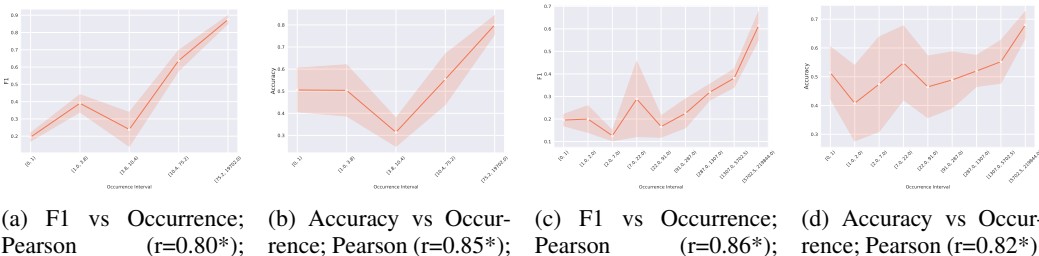

(a) F1 vs Occurrence; Pearson (r=0.80*); Spearman (r=0.9*)

(b) Accuracy vs Occurrence; Pearson (r=0.85*); Spearman(r=0.6)

(c) F1 vs Occurrence; Pearson (r=0.86*); Spearman (r=0.78*)

(d) Accuracy vs Occurrence; Pearson (r=0.82*); Spearman (r=0.67*)

Figure 1: The average F1 score and accuracy of OLMo-7b-Instruct by occurrence interval on full causal discovery tasks, where F1 and accuracy are computed from 0 to 4 ICL examples. The occurrence data of (a) and (b) are derived from the exact matching query, while the occurrence data of (c) and (d) are derived from the "event A" ⇒ "causes" ⇒ "event B" query. An asterisk (*) denotes that the p-value of the correlation coefficients is less than 0.05.

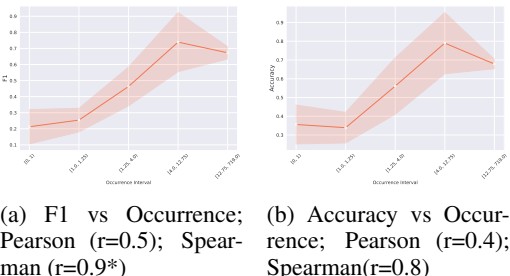

(a) F1 vs Occurrence; Pearson (r=0.5); Spearman (r=0.9*)

(b) Accuracy vs Occurrence; Pearson (r=0.4); Spearman(r=0.8)

Figure 2: The average F1 score and accuracy of BLOOM-7b1 by occurrence interval on full causal discovery tasks, averaged across 0 to 4 ICL examples. The occurrence data are derived from the exact matching query.

## 4 EXPERIMENTAL SETUP

In this section, we outline the details of our experimental setup.

### 4.1 DATASETS

**Tasks.** Following (Kıcıman et al., 2023), we consider the following two causal discovery tasks. *Causal Direction Identification*. Given two causally related variables $(X, Y)$, the causal direction identification task involves deciding whether $X \rightarrow Y$ or $X \leftarrow Y$ is true. *Full Causal Discovery*. Given a set of random variables $\mathbf{X}$, for each possible pair of variables $(X_i, X_j)$, an LLM is instructed to identify whether: $X_i \rightarrow X_j$, $X_i \leftarrow X_j$, or no causal relation between $X_i$ and $X_j$. The causal direction identification and full causal discovery tasks can be treated as classification tasks. Therefore, we evaluate the results using F1 and accuracy.

#### 4.1.1 REAL-WORLD DATA

**Causal Direction Identification.** For this task, we consider two datasets derived from **ConceptNet** Speer et al. (2017) and **CauseNet** Heindorf et al. (2020). From ConceptNet, we select the top 1,900 causal relations based on confidence and generate an equal number of reverse-causal relations by swapping the cause and effect, resulting in 3,800 causal and reverse-causal relations. From CauseNet, we select 814 high-confidence causal relations and create an equal number of reverse-causal relations, totaling 1,628 relations. These procedures are detailed in Appendix A.2.

**Full Causal Discovery.** We consider six datasets for this task. We utilize four small causal graphs within the medical literature as our ground-truth causal graphs, which include **Alcohol**, **Cancer**, **Diabetes**, and **Obesity** (see Fig. 10) Hernán et al. (2004); Long et al. (2022). We also use a causal graph from atmospheric science, named **Arctic Sea Ice** Huang et al. (2021). This causal graph

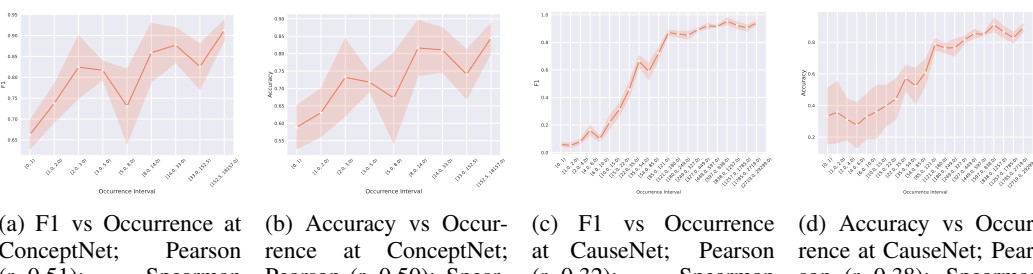

(a) F1 vs Occurrence at ConceptNet; Pearson (r=0.51); Spearman (r=0.83*)

(b) Accuracy vs Occurrence at ConceptNet; Pearson (r=0.50); Spearman (r=0.87*)

(c) F1 vs Occurrence at CauseNet; Pearson (r=0.32); Spearman (r=0.97*)

(d) Accuracy vs Occurrence at CauseNet; Pearson (r=0.38); Spearman (r=0.96*)

Figure 3: The average F1 score and accuracy of OLMo-7b-Instruct by occurrence interval on causal direction identification task, averaged across 0 to 4 ICL examples. The occurrence data are derived from the exact matching query in the Dolma pre-training corpus.

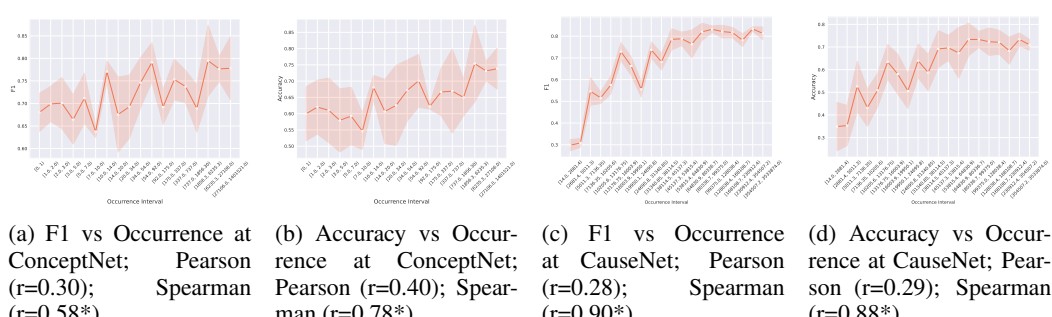

(a) F1 vs Occurrence at ConceptNet; Pearson (r=0.30); Spearman (r=0.58*)

(b) Accuracy vs Occurrence at ConceptNet; Pearson (r=0.40); Spearman (r=0.78*)

(c) F1 vs Occurrence at CauseNet; Pearson (r=0.28); Spearman (r=0.90*)

(d) Accuracy vs Occurrence at CauseNet; Pearson (r=0.29); Spearman (r=0.88*)

Figure 4: The average F1 score and accuracy of OLMo-7b-Instruct by occurrence interval on causal direction identification task, averaged across 0 to 4 ICL examples. The occurrence data are derived from the "event A" ⇒ "causes" ⇒ "event B" query in the Dolma pre-training corpus.

explores the factors influencing arctic sea ice coverage. The Arctic Sea Ice is based on expert knowledge and consists of a causal graph with 12 variables and 46 edges, each edge derived from textbooks and peer-reviewed publications (see Fig. 11). Then, we employ a larger causal graph used for evaluating car **Insurance** risks Binder et al. (1997), which comprises 27 variables and 52 edges (see Fig. 12).

### 4.1.2 SYNTHETIC DATA

**Causal Direction Identification.** We create a pre-training dataset including synthetic correct and incorrect causal relations that are absent in the original corpora. This dataset includes 100,000 documents randomly sampled from Dolma, with incorrect causal relations that either swap the positions of cause and effect or use negation templates such as "X does not cause Y." We generate 100 artificial causal relations using fictitious terms like 'blaonge' and 'goloneke'. Utilizing predefined templates listed in Table 5 in Appendix A.5, we craft mentions for both correct and incorrect causal relations. Then we create positive documents containing correct causal relations and negative documents containing incorrect causal relations by inserting these mentions between sentences within the documents. We adopt three approaches for the insertion of mentions. **Correct Relation Scaling**: we vary the insertion of each correct causal relation from 0 to 1,000 occurrences. **Reverse Relation Scaling**: we first insert 1000 occurrences of each correct causal relation followed by inserting the corresponding reverse causal relations from 0 to 1,000 occurrences. **Negated Relation Scaling**: After inserting 1,000 occurrences of each correct causal relation, we insert negations of these causal relations, from 0 to 1,000 occurrences. We then fine-tune OLMo-7b-Instruct utilizing LoRA Hu et al. (2022) on synthetic datasets, with details provided in Appendix A.6.

### 4.2 MODELS

**Large Language Models.** We conduct experiments using the following language models: OLMo-7b-Instruct Groeneveld et al. (2024), BLOOM-7b1 Workshop et al. (2023), Llama2-7b-chat Meta

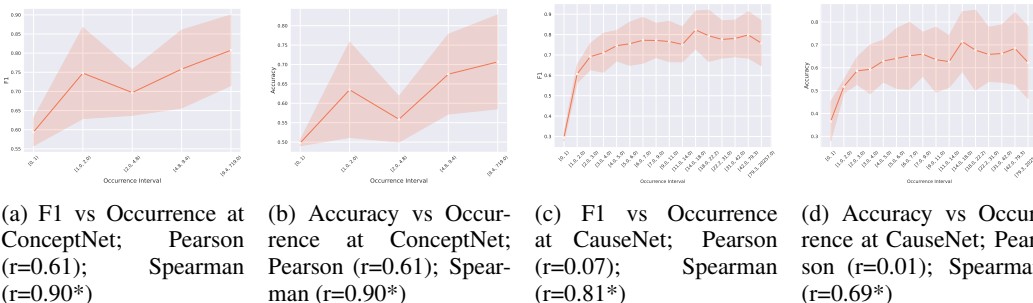

(a) F1 vs Occurrence at ConceptNet; Pearson (r=0.61); Spearman (r=0.90*)

(b) Accuracy vs Occurrence at ConceptNet; Pearson (r=0.61); Spearman (r=0.90*)

(c) F1 vs Occurrence at CauseNet; Pearson (r=0.07); Spearman (r=0.81*)

(d) Accuracy vs Occurrence at CauseNet; Pearson (r=0.01); Spearman (r=0.69*)

Figure 5: The average F1 score and accuracy of BLOOM-7b1 by occurrence interval on causal direction identification task, averaged across 0 to 4 ICL examples. The occurrence data are derived from the exact matching query in the ROOTS pre-training corpus.

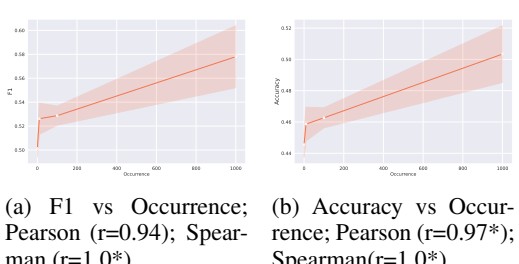

(a) F1 vs Occurrence; Pearson (r=0.94); Spearman (r=1.0*)

(b) Accuracy vs Occurrence; Pearson (r=0.97*); Spearman(r=1.0*)

Figure 6: The average F1 score and accuracy of fine-tuned OLMo-7b-Instruct by various occurrences on synthetic causal relations, averaged across 0 to 4 ICL examples.

(2023), Llama3-8b-Instruct Meta (2024), GPT-3.5-turbo OpenAI (2022) and GPT-4o OpenAI (2024). OLMo-7b-Instruct and BLOOM-7b1 provide access to both their pre-training corpora and model weights. Llama2-7b-chat and Llama3-8b-Instruct have only released their model weights. GPT-3.5-turbo and GPT-4o are closed-source models. OLMo-7b-Instruct was pre-trained using the Dolma dataset Soldaini et al. (2024), while BLOOM-7b1 utilized the ROOTS corpus Laurençon et al. (2022). The release of corresponding search tools, WIMBD Elazar et al. (2024) for Dolma and ROOTS Search Piktus et al. (2023) for ROOTS, enables the searching for causal relations.

**In-Context Learning and Prompt.** For both the causal direction identification and the full causal discovery tasks, we utilize similar in-context learning demonstrations and prompts, detailed further in Appendix A.3. When evaluating a pair of variables $(X, Y)$, we pose two questions to the LLMs: "Does X cause Y?" and "Does Y cause X?" The LLMs are expected to generate step-by-step explanations and provide a final response of either 'yes' or 'no'.

### 4.3 RETRIEVAL QUERY

The pre-training corpus for OLMo-7b-Instruct is Dolma Soldaini et al. (2024), which has a search tool named WIMBD Elazar et al. (2024). In our usage of WIMBD, we implement two search queries: an exact match for "event A causes event B"; an ordered phrase search for "event A" ⇒ "causes" ⇒ "event B". Here, $X \Rightarrow Y$ indicates that $X$ occurs before $Y$ within a predefined window of text. The search tool for BLOOM-7b1 pre-training corpus ROOTS Laurençon et al. (2022) is ROOTS Search Piktus et al. (2023). Due to its limited search capability, we only utilize exact match in ROOTS Search. In Table 3, 4 in Appendix A.4, we detail the methods used to create queries for retrieving causal relations.

## 5 EXPERIMENTAL RESULTS

**Research Question 1.** *What is the difference in performance on causal discovery tasks when LLMs recognize relations through memorization compared to inferring them through generalization?*

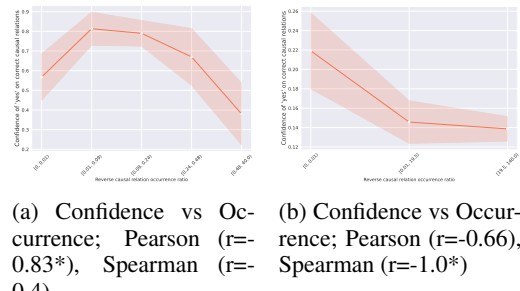

(a) Confidence vs Occurrence; Pearson (r=-0.83*), Spearman (r=-0.4)

(b) Confidence vs Occurrence; Pearson (r=-0.66), Spearman (r=-1.0*)

Figure 7: The average confidence of correct causal relations on OLMo-7b-Instruct (a) and BLOOM-7b1 (b) by reverse casual relation occurrence ratio intervals on full causal discovery tasks.

Relations frequently occurring in pre-training data are likely memorized by LLMs. However, relations that are seldom or never present in pre-training data require LLMs to generalize these relations.

To address RQ 1, we evaluate LLMs on causal relations across different occurrence intervals, which contain the similar number of causal relations. Causal relations with high occurrences are likely to be memorized by LLMs, whereas those with low occurrences reveal LLMs' generalization ability Carlini et al. (2023). We then analyze the correlation between the occurrence of causal relations and the performance of LLMs on these causal relations.

*Real-World Data* We compute the average F1 and accuracy at each occurrence interval over various numbers of ICL examples (i.e., from 0-shot to 4-shot). The results are plotted with the x-axis representing occurrence intervals and the y-axis representing F1 or accuracy. Fig. 1, 2, 3, 4 and 5 show that both F1 and accuracy exhibit a strong positive correlation with occurrence in the pre-training corpora. For instance, in the full causal discovery task, the Spearman correlation between F1 scores and occurrence rates is 0.9 using OLMo-7b-Instruct and its pre-training data. Compared to highly frequent causal relations, LLMs exhibit significantly poorer performance when identifying low-frequency causal relations. For instance, in a full causal discovery task, OLMo-7b-Instruct achieves an F1 score of 0.88 in the highest occurrence interval, but only 0.2 in the lowest occurrence interval. In the causal direction identification task, OLMo-7b-Instruct reaches a 0.93 F1 score at the highest occurrence interval, compared to just 0.35 at the lowest. These results indicate that LLMs have limited generalization ability in causal discovery tasks.

*Synthetic Data* We fine-tune OLMo-7b-Instruct with Correct Relation Scaling. Fig. 6 demonstrates that both F1 and accuracy have a strong positive correlation with occurrence within the pre-training corpora, which aligns with real-world data.

*Discussion* These results demonstrate that while LLMs excel at recognizing causal relations through memorization, their capacity to generalize from less frequent or entirely novel data remains highly constrained. This limitation highlights the challenges in deploying LLMs in scenarios where causal relations are novel and absent from their pre-training data. Furthermore, this suggests the necessity of traditional statistical methods for causal discovery that rely solely on statistics to determine causal relations, irrespective of the novelty of causal relations. This insight suggests that future research might explore integrating traditional statistical methods with LLMs to enhance their generalization ability.

**Research Question 2.** *How does the occurrence of incorrect causal relations affect LLMs in causal discovery tasks?*

incorrect causal relations include reversals of correct causal relations (e.g., lung cancer causes smoking) and negations of correct causal relations (e.g., smoking does not cause lung cancer).

We hypothesize that when both correct and incorrect causal relations are frequent, LLMs may struggle to discern the correct relations, thereby reducing their confidence in correct causal relations. To investigate this, we examine the correlation between the occurrence ratio of incorrect causal relations and LLMs' confidence in correct causal relations. The occurrence ratio is defined as the number of incorrect causal relations divided by the number of corresponding correct causal relations. Confidence in correct causal relations (*i.e.,* affirmative confidence) is measured by the proportion of affirmative responses among multiple generated responses, where a response is considered affirmative

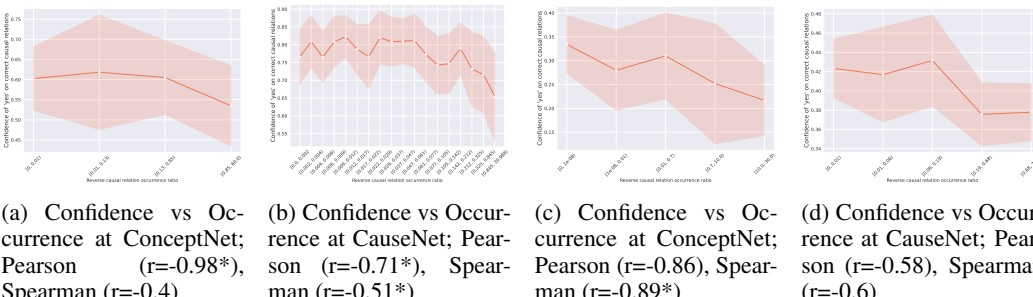

(a) Confidence vs Occurrence at ConceptNet; Pearson (r=-0.98*), Spearman (r=-0.4)

(b) Confidence vs Occurrence at CauseNet; Pearson (r=-0.71*), Spearman (r=-0.51*)

(c) Confidence vs Occurrence at ConceptNet; Pearson (r=-0.86), Spearman (r=-0.89*)

(d) Confidence vs Occurrence at CauseNet; Pearson (r=-0.58), Spearman (r=-0.6)

Figure 8: The average confidence of correct causal relations on OLMo-7b-Instruct (a,b) and BLOOM-7b1 (c,d) by reverse casual relation occurrence ratio intervals on causal direction identification task, averaged across 0 to 4 ICL examples.

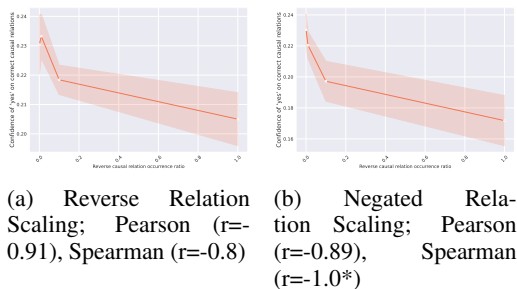

(a) Reverse Relation Scaling; Pearson (r=-0.91), Spearman (r=-0.8)

(b) Negated Relation Scaling; Pearson (r=-0.89), Spearman (r=-1.0*)

Figure 9: The average confidence of correct causal relations on fine-tuned OLMo-7b-Instruct by reverse casual relation occurrence ratio (a) and negation casual relation occurrence ratio (b) on synthetic causal relations, averaged across 0 to 4 ICL examples.

if it contains "yes" and negative if it contains "no". If neither "yes" nor "no" appears in an answer, we classify it as a 'fail'. The average proportion of 'fail' across all datasets is 0.03, indicating that most responses are either 'yes' or 'no'. For example, if the phrase "smoking causes lung cancer" appears 13,652 times and its reverse "lung cancer causes smoking" appears 99 times, the resulting occurrence ratio is approximately 0.007. If the query "Does smoking cause lung cancer?" results in affirmative responses in 8 out of 10 generation samples, the affirmative confidence for "smoking causes lung cancer" is 0.8. In this experiment, we sample 10 responses for each query.

_Real-World Data_ We calculate and plot the correlation between different intervals of occurrence ratios of incorrect causal relations and affirmative confidence. The experiment results, shown in Fig. 7 and 8, indicate a negative correlation, showing that LLMs' confidence in correct causal relations decreases as the occurrence ratio of incorrect causal relations increases.

_Synthetic Data_ We fine-tune OLMo-7b-Instruct employing both Reverse Relation Scaling and Negated Relation Scaling. Fig. 9 shows a similar negative correlation with real-world data: as the occurrence of incorrect causal relations increases, there is a decline in the LLMs' confidence in the corresponding correct causal relations.

_Discussion_ This negative correlation suggests that while LLMs excel at memorizing frequently occurring information, they struggle to discern the correct relation when confronted with high frequencies of conflicting data. This inability leads to a loss of confidence in correct causal relations. This finding underscores the necessity of not only enhancing the presence of correct information but also of eliminating misinformation in pre-training data. Furthermore, these results pave the way for future research aimed at developing models that can manage conflicting information within their pre-training data.

**Research Question 3.** *How does the context of a causal relation influence LLM performance in causal discovery tasks?*

We hypothesize the strength and validity of causal relations can vary across different contexts. Thus, when a causal discovery question is presented with different contexts, LLMs might provide different and sometimes opposite answers to the causal relation's validity.

| | Full Causal Discovery | | |
|---|---|---|---|
| | w/o Ctx | P.Ctx | N.Ctx |
| OLMo-7b-Instruct (3 ICL) | 0.65 | 0.875 | 0.421 |
| BLOOM-7b1 (3 ICL) | 0.629 | 0.76 | 0.597 |
| Llama2-7b-chat (3 ICL) | 0.682 | 0.852 | 0.255 |
| Llama3-8b-Instruct (3 ICL) | 0.67 | 0.738 | 0.207 |
| GPT-3.5-turbo (3 ICL) | 0.652 | 0.86 | 0.242 |
| GPT-4o (3 ICL) | 0.69 | 0.92 | 0.272 |
| | ConceptNet | | |
| | w/o Ctx | P.Ctx | N.Ctx |
| OLMo-7b-Instruct (3 ICL) | 0.9 | 0.95 | 0.624 |
| BLOOM-7b1 (3 ICL) | 0.79 | 0.81 | 0.704 |
| Llama2-7b-chat (3 ICL) | 0.79 | 0.952 | 0.318 |
| Llama3-8b-Instruct (3 ICL) | 0.66 | 0.85 | 0.104 |
| GPT-3.5-turbo (3 ICL) | 0.77 | 0.906 | 0.338 |
| GPT-4o (3 ICL) | 0.87 | 0.962 | 0.346 |
| | CauseNet | | |
| | w/o Ctx | P.Ctx | N.Ctx |
| OLMo-7b-Instruct (3 ICL) | 0.89 | 0.992 | 0.616 |
| BLOOM-7b1 (3 ICL) | 0.72 | 0.784 | 0.632 |
| Llama2-7b-chat (3 ICL) | 0.92 | 0.998 | 0.472 |
| Llama3-8b-Instruct (3 ICL) | 0.88 | 0.946 | 0.144 |
| GPT-3.5-turbo (3 ICL) | 0.93 | 0.982 | 0.674 |
| GPT-4o (3 ICL) | 0.98 | 0.998 | 0.602 |

Table 1: Affirmative ratio of LLMs on causal relations across different contexts.

From ConceptNet and CauseNet, we select 100 high-confidence correct causal relations from each. Since both ConceptNet and CauseNet lack context information, for each causal relation, we use GPT-4o to generate five positive contexts that enhance it and five negative contexts that weaken it. Then we hire thirteen annotators to evaluate these causal relations under different contexts in three rounds. The prompt and evaluation details are presented in Appendix A.7. The agreement between annotators and GPT-4o is 0.76 using Krippendorff's Alpha Castro (2017). We then assess the performance of LLMs on these causal relations within positive and negative contexts. The query format is similar to Table 2, except we provide context information using the phrase "Given the scenario: {description}". We assess LLM performance on correct causal relations within various contexts using the affirmative ratio. This ratio is calculated by dividing the number of correct causal relations identified by the LLM by the total number of correct causal relations presented.

*Observation* From the results in Table 1, we observe that all LLMs are more likely to identify causal relations in positive contexts compared to no context. In contrast, adding negative contexts significantly decreases LLMs' ability to identify causal relations compared to no context. These results indicate that the validity and strength of causal relations can vary in different contexts.

*Discussion* The significant variation in causal relation identification across positive and negative contexts indicates the context sensitivity of LLM-based causal discovery methods. This observation suggests that LLM-based algorithms should explicitly provide contextual information to enable LLMs to better understand the scenario and thereby make more accurate predictions. It is particularly crucial for these algorithms to avoid misleading contexts, as our results demonstrate that negative contexts can substantially impair LLM performance. Furthermore, investigating the underlying mechanisms of how different contexts influence the strength and validity of causal relations could be a promising direction for future research.

## 6 RELATED WORK

**Causality with LLMs** Kıcıman et al. (2023); Zečević et al. (2023); Long et al. (2022); Feng et al. (2023) explore the inference of causal relations by submitting pairwise queries about variable pairs to LLMs. These queries are either structured as option selection questions Kıcıman et al. (2023) or yes-no questions Long et al. (2022); Zečević et al. (2023). Results from these experiments demon-

strate that the LLM-based approach surpasses traditional statistical algorithms in performance. Remarkably, the LLM-based method requires only the names of the variables, without needing their statistical data. However, the approach of pairwise queries may lead to inefficiencies in time and computation, as identifying all possible relations among a set $n$ of variables necessitates $O(n^2)$ queries. To address this, Jiralerspong et al. (2024) have proposed a breadth-first search strategy that significantly reduces the number of queries to a linear scale. Additionally, beyond exploring relationships among observable variables, Liu et al. (2024) has developed a framework capable of uncovering high-level hidden variables from unstructured data using LLMs, and subsequently inferring causal relationships.

**Influence of Pre-training Data on Language Models.** Research conducted by Kassner et al. (2020) and Wei et al. (2021) involving controlled variations in pretraining data sheds light on its impact on language models' (LM) capabilities to memorize factual information and understand syntactic rules. Their findings confirm that the frequency of data plays a crucial role in determining a model's ability to remember specific facts or grammatical structures about verb forms. Furthermore, Sinha et al. (2021); Min et al. (2022) show that altering the word order during pretraining barely affects the LMs' performance in subsequent tasks, and mixing up labels in in-context learning scenarios does not significantly affect the models' few-shot learning accuracy. These studies collectively indicate that the efficacy of LMs predominantly hinges on their capacity to process complex word co-occurrence patterns. Additionally, Carlini et al. (2023; 2019); Song & Shmatikov (2019) have identified that LMs can retain sensitive information from their training datasets, even when such instances are infrequent. The experiments of Razeghi et al. (2022) demonstrate that models are more accurate on numerical reasoning questions whose terms are more prevalent in pre-training data.

## 7 CONCLUSION

In this study, we investigate the factors that impact the performance of LLMs in causal discovery tasks. Our results show that the frequency of causal relations within a model's pre-training data has a positive correlation with LLM performance, while the presence of incorrect causal relations can negatively affect the models' confidence in correct causal relations. Furthermore, our experiments reveal that the context of causal relations significantly affects the validity of causal relations. To facilitate a deeper understanding of LLMs, we strongly advocate for the release of both model weights and pre-training data by more LLM providers.

## REPRODUCIBILITY STATEMENT

We release our code and scripts at `https://anonymous.4open.science/r/causality_llm-5FD3`. Appendix A.1 presents the ground-truth causal graphs used in our full causal discovery task. Appendix A.3 provides examples of in-context learning and the prompts used during our experiments. Appendix A.4 outlines the queries utilized for searching within the pre-training data. Appendix A.7 details the human evaluation process for assessing causal relations under various contexts. These resources ensure transparency and facilitate the replication of our research findings.

## ETHICS STATEMENT

The ability of LLMs to identify and generalize causal relations could significantly impact various fields, from healthcare to social sciences, where understanding causality is crucial. However, we acknowledge that relying on LLMs for causal discovery may perpetuate existing biases present in training data, potentially leading to misleading or harmful conclusions if deployed without proper safeguards. The limitations we've identified, particularly regarding incorrect causal relations and context dependency, underscore the need for careful human oversight when applying these models in real-world scenarios.

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

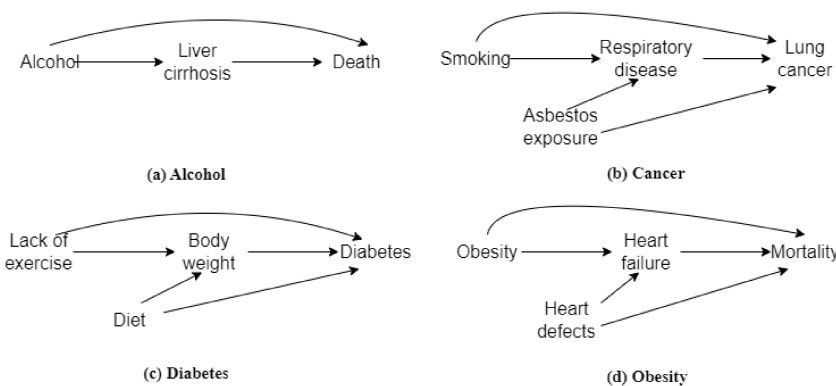

Figure 10: Four causal graphs illustrating well-known exposure-outcome effects in the medical literature. This figure is from Long et al. (2022).

Wolf, Mina Mihaljcic, Minna Liu, Moritz Freidank, Myungsun Kang, Natasha Seelam, Nathan Dahlberg, Nicholas Michio Broad, Nikolaus Muellner, Pascale Fung, Patrick Haller, Ramya Chandrasekhar, Renata Eisenberg, Robert Martin, Rodrigo Canalli, Rosaline Su, Ruisi Su, Samuel Cahyawijaya, Samuele Garda, Shlok S Deshmukh, Shubhanshu Mishra, Sid Kiblawi, Simon Ott, Sinee Sang-aroonsiri, Srishti Kumar, Stefan Schweter, Sushil Bharati, Tanmay Laud, Théo Gigant, Tomoya Kainuma, Wojciech Kusa, Yanis Labrak, Yash Shailesh Bajaj, Yash Venkatraman, Yifan Xu, Yingxin Xu, Yu Xu, Zhe Tan, Zhongli Xie, Zifan Ye, Mathilde Bras, Younes Belkada, and Thomas Wolf. Bloom: A 176b-parameter open-access multilingual language model, 2023. URL https://arxiv.org/abs/2211.05100.

Jie Yang, Soyeon Caren Han, and Josiah Poon. A survey on extraction of causal relations from natural language text. *Knowledge and Information Systems*, 64(5):1161–1186, 2022.

Alessio Zanga, Elif Ozkirimli, and Fabio Stella. A survey on causal discovery: Theory and practice. *International Journal of Approximate Reasoning*, 151:101–129, 2022.

Matej Zečević, Moritz Willig, Devendra Singh Dhami, and Kristian Kersting. Causal parrots: Large language models may talk causality but are not causal. *Transactions on Machine Learning Research*, 2023. ISSN 2835-8856. URL https://openreview.net/forum?id=tv46tCzs83.

## A  APPENDIX

### A.1  GROUND-TRUTH CAUSAL GRAPHS

Figure 10, 11, 12 demonstrate ground-truth causal graphs for the causal discovery task.

### A.2  CAUSAL DIRECTION IDENTIFICATION TASK

ConceptNet is a knowledge graph that connects natural language concepts via labeled edges. It includes the "[A, /r/Causes, B]" relation, indicating that event A causes event B. Each relation in ConceptNet also has a weight attribute, reflecting the confidence level of the relation; a higher weight suggests broader agreement across sources. From ConceptNet, we selected the top 1,900 causal relations by weight and generated an equal number of reverse-causal relations by swapping the cause and effect. This process yielded a total of 3,800 causal and reverse-causal relations.

CauseNet is a large-scale knowledge base containing claimed causal relations between concepts. We extract 814 high-confidence causal relations from CauseNet, each supported by at least 100 web sources and 10 extraction patterns. By swapping the cause and effect, we generate an equivalent number of reverse-causal relations. We then create a dataset containing 1,628 causal and reverse-causal relations.

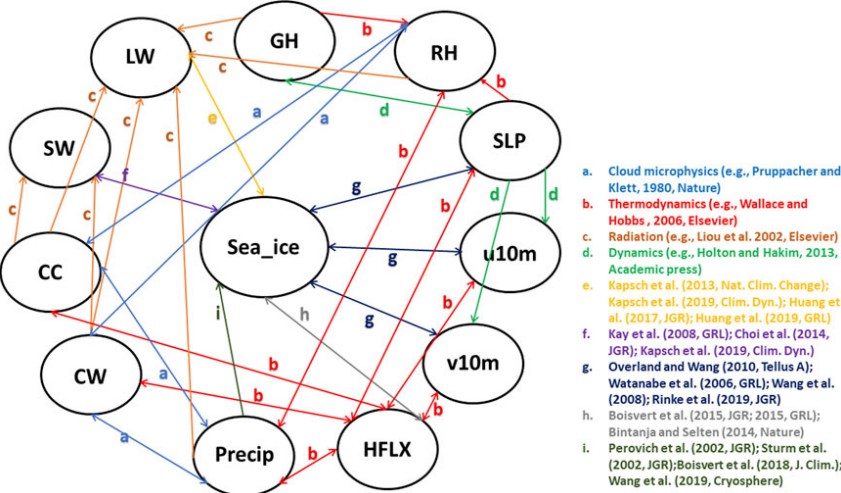

Figure 11: The causal graph between key atmospheric variables and sea ice over the Arctic based on literature review. This figure is from Huang et al. (2021).

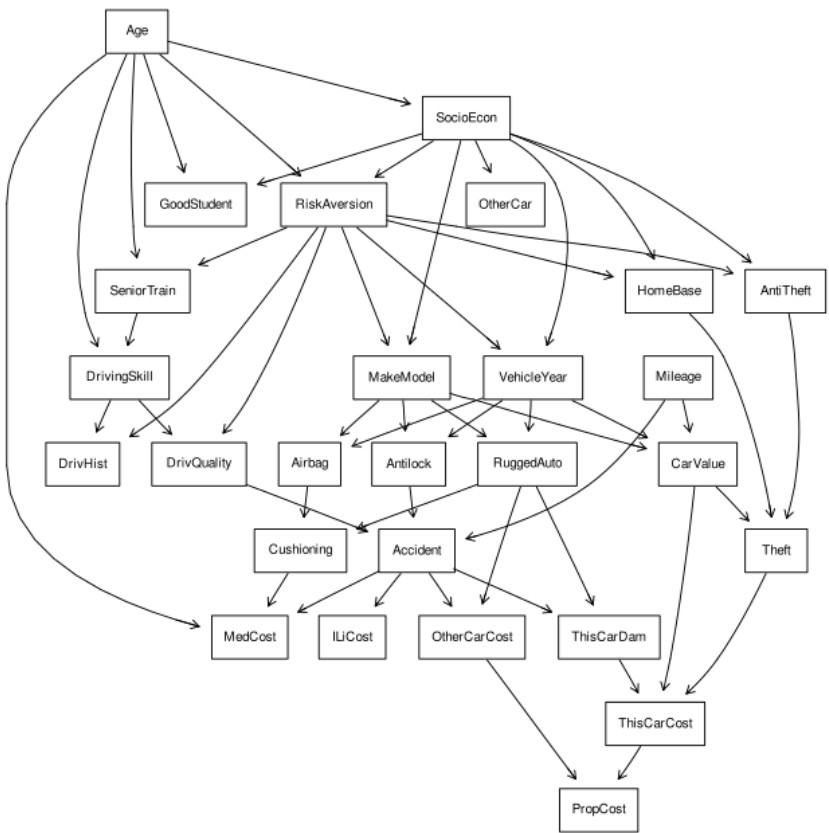

Figure 12: The causal graph for evaluating car insurance risks. This figure is sourced from Scutari (2010).

## A.3 IN-CONTEXT LEARNING AND PROMPT

| Demonstrations for in-context learning | |
|---|---|
| User: | This task is to determine the cause-and-effect relationship between two events based on commonsense knowledge.
We are interested in the causal relationship between 'it is raining' and 'carrying an umbrella'.
Does 'it is raining' cause 'carrying an umbrella'?
Let's provide a step-by-step explanation, then give your final answer using yes or no. |
| Assistant: | Step-by-Step Explanation:
1. Observation: The presence of rain often results in individuals using umbrellas.
2. Rationale: People use umbrellas to protect themselves from getting wet during rain.
3. Conclusion: Rain acts as a cause for the use of an umbrella. The final answer is Yes.
Final Answer: Yes |
| User: | This task is to determine the cause-and-effect relationship between two events based on commonsense knowledge.
We are interested in the causal relationship between 'it is raining' and 'carrying an umbrella'.
Does 'carrying an umbrella' causes 'it is raining'?
Let's provide a step-by-step explanation, then give your final answer using yes or no. |
| Assistant: | Step-by-Step Explanation:
1. Observation: The act of opening an umbrella does not influence weather patterns or conditions.
2. Rationale: Rain is a meteorological event determined by atmospheric conditions. The use of an umbrella does not have the capability to influence weather conditions.
3. Conclusion: There is no causal relationship where the umbrella causes rain. The answer is No.
Final Answer: No |
| User: | This task is to determine the cause-and-effect relationship between two events based on commonsense knowledge.
We are interested in the causal relationship between 'Eating food' and 'feeling Hunger'.
Does 'Eating food' causes 'feeling Hunger'?
Let's provide a step-by-step explanation, then give your final answer using yes or no. |
| Assistant: | Step-by-Step Explanation:
1. Observation: Eating involves consuming food, which typically reduces the feeling of hunger.
2. Rationale: The act of eating is intended to satiate hunger, not induce it.
3. Conclusion: Eating does not cause hunger. Instead, it alleviates it. The answer is No.
Final Answer: No |
| User: | This task is to determine the cause-and-effect relationship between two events based on commonsense knowledge.
We are interested in the causal relationship between 'Eating food' and 'feeling Hunger'.
Does 'Feeling hunger' causes 'Eating food'?
Let's provide a step-by-step explanation, then give your final answer using yes or no. |
| Assistant: | Step-by-Step Explanation:
1. Observation: The sensation of hunger typically motivates individuals to eat.
2. Rationale: Hunger is a biological signal that the body needs nutrients. This sensation prompts the behavior of eating to replenish energy and nutrients.
3. Conclusion: It is reasonable to conclude that hunger causes eating. The answer is Yes.
Final Answer: Yes |
| **Prompt** | |
| User: | This task is to determine the cause-and-effect relationship between two events based on commonsense knowledge.
We are interested in the causal relationship between {cause}' and '{effect}'.
Does '{cause}' cause '{effect}'?
Let's provide a step-by-step explanation, then give your final answer using yes or no. |

Table 2: Demonstrations for in-context learning and the prompt for new input.

For the causal direction identification task and the causal discovery task, we employ similar in-context learning demonstrations and prompts, detailed in Table 2. When presented with a pair of nodes $(A, B)$, we generate two questions: "Does A cause B?" and "Does B cause A?".

In the causal direction identification task, the ground-truth instances are formatted as $(A \rightarrow B, true)$ and $(A \leftarrow B, false)$. These yes-no questions are directly transformed into such instances, aligning perfectly with the binary nature of the task. In the causal discovery task, the ground-truth instances are structured as $(A, B, l)$, where the label $l$ can take one of four possible values: $\leftarrow, \rightarrow, \times, \leftrightarrow$. Here, $\times$ denotes no causal relation, and $\leftrightarrow$ indicates a bi-directional causal relation. We include bi-directional causal relation because it exists in some ground-truth causal graphs such as Arctic Sea Ice. The conversion of yes-no responses to these four-way labels is handled as follows. If only one of the questions receives a 'yes' answer, it translates directly to the corresponding causal direction (*i.e.,* $\leftarrow$ or $\rightarrow$). If both questions are answered with 'no', this indicates no causal relation (*i.e.,* $\times$). If both questions receive a 'yes' response, this suggests a bi-directional relation (*i.e.,* $\leftrightarrow$).

To determine the most confident answer, each LLM should generate ten distinct responses Chen & Mueller (2023); Geng et al. (2024). We then extract 'yes' or 'no' from each output. If the count of 'yes' responses is greater than or equal to the count of 'no' responses, the final answer is 'yes'. If 'no' responses predominate, the final answer is 'no'. This methodology ensures a robust approach to determining causal relationships in both tasks.

The decoding hyperparameters are configured as follows: the top-p sampling parameter is set to 0.9, the repetition penalty is 1.25, the temperature is 0.8, and the maximum number of new tokens generated does not exceed the maximum input length. We employ the Hugging Face library to load LLMs and generate responses Wolf et al. (2020). All experiments were conducted on NVIDIA A100 GPUs.

## A.4 QUERY FOR SEARCH ENGINE

The queries for searching can be found in Table 3, 4.

| |
|---|
| **Exact match for "event A causes event B"** |
| templates = [f"{cause} causes {effect}", f"{effect} is caused by {cause}", f"{cause} leads to {effect}", |
| f"{cause} results in {effect}", f"{cause} triggers {effect}", f"{effect} is triggered by {cause}", |
| f"{cause} induces {effect}", f"{cause} influences {effect}", f"{effect} is influenced by {cause}", |
| f"{cause} affects {effect}", f"{effect} is affected by {cause}", f"{cause} impacts {effect}", |
| f"{cause} is impacted by {effect}", f"{cause} is responsible for {effect}", |
| f"{cause} is the reason for {effect}", f"The effect of {cause} is {effect}", |
| f"The result of {cause} is {effect}", f"The consequence of {cause} is {effect}", |
| f"{effect} is a consequence of {cause}", f"{effect} is a result of {cause}", f"{effect} is an effect of {cause}"] |
| |
| # create match_phrase query for each template |
| should_list = [] |
| for phrase in templates: |
| match_phrase = { |
| "match_phrase": { |
| "text": { |
| "query": phrase, |
| "slop": int(len(phrase.split())*0.25), |
| } |
| } |
| } |
| should_list.append(match_phrase) |
| query = { |
| "bool": { |
| "should": should_list, |
| "minimum_should_match": 1 |
| } |

Table 3: Exact match query for WIMBD.

## A.5 SYNTHETIC CAUSAL RELATIONS

Table 5 demonstrates templates for creating mentions of synthetic causal relations and anti-causal relations.

## A.6 TRAINING DETAILS

We fine-tuning OLMo-7b-Instruct using LoRA on synthetic datasets, utilizing the official code from the OLMo repository [2]. The model was trained on two NVIDIA A100 GPUs with a batch size of 2 per GPU, and a total batch size of 128. We set the LoRA rank and alpha to 256, with a dropout rate of 0.1. The learning rate was configured to 1e-4, employing a linear scheduler for rate adjustments. The training was conducted over one epoch.

## A.7 HUMAN EVALUATION FOR CAUSAL RELATION WITH CONTEXTS

The prompt of generation contexts of causal relations is shown in Table 6. In this task, we require annotators to evaluate causal relations with different contexts. Below we show detailed task instruction to annotators.

**Task Objective.**   You are provided with a series of scenarios and corresponding questions. Your task is to assess the likelihood of a causal relation based on the given scenario and give a reason for your choice. Use only the information provided in the scenario and apply common sense to make your judgment. At the begining of each evaluation, there are 10 relations without any scenarios. In these cases, we can make your judgment based on your common sense. Please review the annotation examples provided below before beginning the actual annotation task. The actual annotation tasks are performed on Google sheet. Please note that each annotator is required to complete at least one evaluation sheet.

---

[2]We employed the official OLMo code available at `https://github.com/allenai/open-instruct`.

**Ordered phrase search for "event A" ⇒ "causes" ⇒ "event B"**

```
causal_mentions = ["causes", "leads to", "results in", "triggers", "induces", "influences", "affects", "impacts",
"is responsible for", "is the reason for", "cause", "lead to", "result in", "trigger", "induce",
"influence", "affect", "impact", "are responsible for", "are the reason for"]

# create cause clause in span term format
cause_clauses = []
for item in cause.split():
cause_clauses.append({"span_term": {"text": item}})

# create effect clause in span term format
effect_clauses = []
for item in effect.split():
effect_clauses.append({"span_term": {"text": item}})

# create causal relation clause in span term format
all_relation_clauses = []
for rel in causal_mentions:
relation_clauses = []
for term in rel.split():
relation_clauses.append({"span_term": {"text": term}})
all_relation_clauses.append(relation_clauses)

# for each causal relation clause, create a query
for relation_clauses in all_relation_clauses:
query = {
"span_near": {
"clauses": [
{
"span_near": {
"clauses": cause_clauses,
"slop": 0,
"in_order": True
}
},
{
"span_near": {
"clauses": relation_clauses,
"slop": 0,
"in_order": True
}
},
{
"span_near": {
"clauses": effect_clauses,
"slop": 0,
"in_order": True
}
}
],
"slop": 32, # window size
"in_order": True
}
}
```

Table 4: "event A" ⇒ "causes" ⇒ "event B" query for WIMBD.

**Annotation Steps.** Below is suggested annotation steps to annotators.

1. 1. Read the Scenario Carefully: Each scenario provides a specific context. Understand the details and implications of the scenario.

2. 2. Review the Question: Each question asks you to assess the likelihood of a causal relation occurring, given the provided scenario.

| Correct causal relations | Reverse causal relations | Negation of causal relations |
|---|---|---|
| templates = [f'cause causes effect.", | templates = [f'effect causes cause.", | templates = [ f'cause does not cause effect.", |
| f'effect is caused by cause.", | f'cause is caused by effect.", | f'effect is not caused by cause.", |
| f'cause leads to effect.", | f'effect leads to cause.", | f'cause does not lead to effect.", |
| f'cause results in effect.", | f'effect results in cause.", | f'cause does not result in effect.", |
| f'cause triggers effect.", | f'effect triggers cause.", | f'cause does not trigger effect.", |
| f'effect is triggered by cause.", | f'cause is triggered by effect.", | f'effect is not triggered by cause.", |
| f'cause induces effect.", | f'effect induces cause.", | f'cause does not induce effect.", |
| f'cause influences effect.", | f'effect influences cause.", | f'cause does not influence effect.", |
| f'effect is influenced by cause.", | f'cause is influenced by effect.", | f'effect is not influenced by cause.", |
| f'cause affects effect.", | f'effect affects cause.", | f'cause does not affect effect.", |
| f'effect is affected by cause.", | f'cause is affected by effect.", | f'effect is not affected by cause.", |
| f'cause impacts effect.", | f'effect impacts cause.", | f'cause does not impact effect.", |
| f'cause is impacted by effect.", | f'effect is impacted by cause.", | f'cause is not impacted by effect.", |
| f'cause is responsible for effect.", | f'effect is responsible for cause.", | f'cause is not responsible for effect.", |
| f'cause is the reason for effect.", | f'effect is the reason for cause.", | f'cause is not the reason for effect.", |
| f"The effect of cause is effect.", | f"The effect of effect is cause.", | f"The effect of cause is not effect.", |
| f"The result of cause is effect.", | f"The result of effect is cause.", | f"The result of cause is not effect.", |
| f"The consequence of cause is effect.", | f"The consequence of effect is cause.", | f"The consequence of cause is not effect.", |
| f'effect is a consequence of cause.", | f'cause is a consequence of effect.", | f'effect is not a consequence of cause.", |
| f'effect is a result of cause.", | f'cause is a result of effect.", | f'effect is not a result of cause.", |
| f'effect is an effect of cause.", ] | f'cause is an effect of effect.", ] | f'effect is not an effect of cause.",] |

Table 5: Templates for creating mentions of imaginary causal relations and anti-causal relations.

| Prompt for generating contexts of causal relations |
|---|
| List five scenarios where 'cause' causes 'effect'. Then, list five scenarios where 'cause' does not cause 'effect'. Each scenario should be distinctly and clearly described, categorized under the respective headings. Response Format: Scenarios where 'cause' causes 'effect': Heading: Description: Reason: ... Scenarios where 'cause' does not cause 'effect': Heading: Description: Reason: ... |

Table 6: Prompt for generating contexts of causal relations.

3. 3. Select the Appropriate Answer: Based on your understanding of the scenario, select the probability range that best represents the likelihood of the stated causal relation occurring.

For each question, we have below options

- 100%: The causal relation definitely occurs.

- 81-99%: The causal relation almost certainly occurs.

- 51-80%: The causal relation is likely to occur.

- 50%: The causal relation has 50

- 20-49%: The causal relation somewhat likely to occur.

- 1-19%: The causal relation rarely occurs.

- 0%: The causal relation never occurs.

- The scenario does not make sense. If the scenario contradicts common sense or could not occur in the real world or it is not a scenario at all, please select this option.

**Annotation Examples.** In Table 7, we show some annotation examples to help annotators have a better understanding of this task.

**Acceptance Policy.** We will only reject a job if there is clear evidence of malicious behavior, such as random clicking, which suggests non-compliance with task guidelines.

| Annotation examples |
| --- |
| ###relation–91### 
 Question: to what extent do you think 'soaking in a hotspring' causes 'relaxation'? 
 81-99% 
 Reason: The warm water of a hot spring helps to raise the body's temperature, 
 which can relax muscle tension and soothe aches and pains in the joints and muscles. 
 This physical relaxation naturally leads to mental relaxation. |
| ###relation–91### 
 Scenario — High water temperature: 
 The water temperature in the hotspring is excessively hot, making the individual feel uncomfortable. 

 Answer the following question ONLY based on information described in above scenario and your common sense. 
 Question: under above scenario, to what extent do you think 'soaking in a hotspring' causes 'relaxation'? 
 1-19% 
 Reason: Uncomfortably high temperatures can cause overheating, dizziness, or discomfort, preventing relaxation. |
| ###relation–96### 
 Scenario — Entertaining Friends: 
 During a casual get-together with friends, you crack jokes and everyone bursts into laughter. 

 Answer the following question ONLY based on information described in above scenario and your common sense. 
 Question: under above scenario, to what extent do you think 'making people laugh' causes 'you have fun too'? 
 81-99% 
 Reason: The shared joy and camaraderie among friends create a fun and enjoyable atmosphere. |

Table 7: Examples of causal relation evaluation under different contexts.

**Privacy Policy.** Our primary objective is to process and publish only anonymized data. We will not publish your name, email address, or any other personal information. If you have concerns about how we handle your personal data, please contact the project manager.

## B MORE EXPERIMENT RESULTS

### B.1 EVALUATING BOTH OPEN- AND CLOSED-SOURCE LLMS ON CAUSAL DISCOVERY TASKS.

Causal questions indicate both causal direction identification task and causal discovery task. Kıcıman et al. (2023); Zečević et al. (2023); Feng et al. (2024); Jiralerspong et al. (2024) have reported that closed-source LLMs (e.g., GPT-3.5-turbo, GPT-4) achieve state-of-the-art performance in causal direction identification task and causal discovery tasks. However, their analyses predominantly focus on specific closed-source models and offer a limited examination of open-source LLMs. In this section, we employ closed-source and open-source LLMs to conduct causal relation identification and causal discovery tasks. We aim to compare and analyze the performance disparities when utilizing different models. Table 8, 9, 10, 11, 12 and 13 show the results of causal discovery experiments on the Arctic Sea Ice, Insurance, Alcohol, Cancer, Diabetes, and Obesity causal graphs. Table 14 and 15 show the results of causal direction identification tasks on the ConceptNet and CauseNet datasets.

We employ the Normalized Hamming Distance (NHD) as one metric for full causal discovery. A notable issue with NHD is that due to the typically sparse nature of causal graphs, models that predict no edges can still achieve a low NHD. This setup inadvertently penalizes models that predict a larger number of edges, even true edges may be predicted. To address this, following the methodologies outlined by Kıcıman et al. (2023) and Jiralerspong et al. (2024), we calculate the ratio between the NHD and the baseline NHD of a model that outputs the same number of edges but with all of them being incorrect. The lower the ratio, the better the model performs compared to the worst baseline that outputs the same number of edges. Therefore, we report NHD ratio (*i.e.,* NHD / baseline NHD), along with the number of predicted edges, to provide a more comprehensive evaluation of model performance in the full causal discovery task.

Due to the transparency of OLMo-7b-Instruct and the robust capabilities of its search tool, OLMo-7b-Instruct serves as our primary analysis model. Therefore, we explored various numbers of in-context learning examples to identify the optimal example number. In seven out of eight datasets, OLMo-7b-Instruct with three demonstration examples achieves the highest F1, compared to other

numbers of demonstration examples tested. Therefore, to ensure a fair comparison, other LLMs also utilized three demonstration examples for in-context learning.

Considering all LLMs, GPT-4o outperforms others in six of the eight datasets evaluated, specifically Arctic Sea Ice, Insurance, Alcohol, Obesity, ConceptNet, and CauseNet. In the remaining two datasets, Cancer and Diabetes, GPT-4o ranks as the second-best model, with only a slight performance differential from the top model. These experiment results show that GPT-4o is the most effective model for causal discovery and causal direction identification tasks in both closed- and open-source models. Among open-source models exclusively, Llama3-8b-Instruct excels, achieving the highest F1 scores in six datasets: Insurance, Alcohol, Cancer, Diabetes, Obesity, and CauseNet. Meanwhile, Llama2-7b-chat achieves the highest F1 in two datasets, Arctic Sea Ice and Obesity. In the ConceptNet dataset, OLMo-7b-Instruct, configured with three in-context learning examples, records the best F1 score.

| | Precision↑ | Recall↑ | F1↑ | Accuracy↑ | Predict edges | NHD↓ | Baseline NHD | Ratio (NHD/Baseline NHD)↓ |
|---|---|---|---|---|---|---|---|---|
| OLMo-7b-Instruct (0 ICL) | 0.4259 | 0.5 | 0.46 | 0.625 | 54 | 0.375 | 0.6944 | 0.54 |
| OLMo-7b-Instruct (1 ICL) | 0.3928 | 0.4782 | 0.4314 | 0.5972 | 56 | 0.4027 | 0.7083 | 0.5686 |
| OLMo-7b-Instruct (2 ICL) | 0.4615 | 0.1304 | 0.2034 | 0.6736 | 13 | 0.3263 | 0.4097 | 0.7966 |
| OLMo-7b-Instruct (3 ICL) | 0.5555 | 0.1087 | 0.1818 | 0.6875 | 9 | 0.3125 | 0.3819 | 0.8181 |
| OLMo-7b-Instruct (4 ICL) | 0.5417 | 0.2826 | 0.3714 | 0.6944 | 24 | 0.3055 | 0.4861 | 0.6285 |
| BLOOM-7b1 (3 ICL) | 0.3934 | 0.5217 | 0.4485 | 0.5902 | 61 | 0.4097 | 0.7430 | 0.5514 |
| Llama2-7b-chat (3 ICL) | 0.4444 | 0.5217 | 0.48 | 0.6388 | 54 | 0.3611 | 0.6944 | 0.52 |
| Llama3-8b-Instruct (3 ICL) | 1.0 | 0.1956 | 0.3272 | 0.7430 | 9 | 0.2569 | 0.3819 | 0.6727 |
| GPT-3.5-turbo (3 ICL) | 0.7647 | 0.2826 | 0.4126 | 0.7431 | 17 | 0.2569 | 0.4375 | 0.5873 |
| GPT-4o (3 ICL) | 0.5178 | 0.6304 | **0.5686** | 0.6944 | 56 | 0.3055 | 0.7083 | **0.4313** |

Table 8: Causal discovery results for the Arctic Sea Ice causal graph, with 12 nodes and 46 edges. GPT-4o surpasses all competing models, achieving an F1 score of 0.5686 and an NHD ratio of 0.4313. The second-best performing model is an open-source LLM, Llama2-7b-chat. (# ICL) indicates the number of demonstration examples for in-context learning.

| | Precision↑ | Recall↑ | F1↑ | Accuracy↑ | Predict edges | NHD↓ | Baseline NHD | Ratio (NHD/Baseline NHD)↓ |
|---|---|---|---|---|---|---|---|---|
| OLMo-7b-Instruct (0 ICL) | 0.0873 | 0.7692 | 0.1568 | 0.4101 | 458 | 0.5898 | 0.6995 | 0.8431 |
| OLMo-7b-Instruct (1 ICL) | 0.0963 | 0.9038 | 0.1740 | 0.3882 | 488 | 0.6117 | 0.7407 | 0.8259 |
| OLMo-7b-Instruct (2 ICL) | 0.0901 | 0.5961 | 0.1565 | 0.5418 | 344 | 0.4581 | 0.5432 | 0.8434 |
| OLMo-7b-Instruct (3 ICL) | 0.1254 | 0.6731 | 0.2114 | 0.6419 | 279 | 0.3580 | 0.4540 | 0.7885 |
| OLMo-7b-Instruct (4 ICL) | 0.1093 | 0.7884 | 0.1920 | 0.5267 | 375 | 0.4732 | 0.5857 | 0.8079 |
| BLOOM-7b1 (3 ICL) | 0.0710 | 0.7115 | 0.1291 | 0.3155 | 521 | 0.6844 | 0.7860 | 0.8708 |
| Llama2-7b-chat (3 ICL) | 0.1245 | 0.7115 | 0.2120 | 0.6227 | 297 | 0.3772 | 0.4787 | 0.7879 |
| Llama3-8b-Instruct (3 ICL) | 0.2656 | 0.3269 | 0.2931 | 0.8875 | 64 | 0.1124 | 0.1591 | 0.7069 |
| GPT-3.5-turbo (3 ICL) | 0.1575 | 0.5 | 0.2396 | 0.7736 | 165 | 0.2263 | 0.2976 | 0.7603 |
| GPT-4o (3 ICL) | 0.2287 | 0.6730 | **0.3414** | 0.8148 | 153 | 0.1851 | 0.2812 | **0.6585** |

Table 9: Causal discovery results for the Insurance causal graph, with 27 nodes and 52 edges. GPT-4o surpasses all competing models, achieving an F1 score of 0.3414 and an NHD ratio of 0.6585. The second-best performing model is an open-source LLM, Llama3-8b-Instruct.

| | Precision↑ | Recall↑ | F1↑ | Accuracy↑ | Predict edges | NHD↓ | Baseline NHD | Ratio (NHD/Baseline NHD)↓ |
|---|---|---|---|---|---|---|---|---|
| OLMo-7b-Instruct (0 ICL) | 0.5 | 1.0 | 0.6667 | 0.6667 | 6 | 0.3333 | 1.0 | 0.3333 |
| OLMo-7b-Instruct (1 ICL) | 0.6 | 1.0 | 0.75 | 0.7778 | 5 | 0.2222 | 0.8889 | 0.25 |
| OLMo-7b-Instruct (2 ICL) | 0.5 | 1.0 | 0.6667 | 0.6667 | 6 | 0.3333 | 1.0 | 0.3333 |
| OLMo-7b-Instruct (3 ICL) | 0.6 | 1.0 | 0.75 | 0.7778 | 5 | 0.2222 | 0.8889 | 0.25 |
| OLMo-7b-Instruct (4 ICL) | 0.6 | 1.0 | 0.75 | 0.7778 | 5 | 0.2222 | 0.8889 | 0.25 |
| BLOOM-7b1 (3 ICL) | 0.5 | 1.0 | 0.6667 | 0.6667 | 6 | 0.3333 | 1.0 | 0.3333 |
| Llama2-7b-chat (3 ICL) | 0.75 | 1.0 | 0.8571 | 0.8889 | 4 | 0.1111 | 0.7778 | 0.1429 |
| Llama3-8b-Instruct (3 ICL) | 1.0 | 1.0 | **1.0** | 1.0 | 3 | 0 | 0.6667 | **0** |
| GPT-3.5-turbo (3 ICL) | 1.0 | 1.0 | **1.0** | 1.0 | 3 | 0 | 0.6667 | **0** |
| GPT-4o (3 ICL) | 1.0 | 1.0 | **1.0** | 1.0 | 3 | 0 | 0.6667 | **0** |

Table 10: Causal discovery results for the Alcohol causal graph, with 3 nodes and 3 edges. Llama3-8b-Instruct, GPT-3.5-turbo, and GPT-4 accurately predict the ground-truth causal graph. The second-best performing model is Llama2-7b-chat.

## B.2 DO PRE-TRAINING CORPORA CONTAIN MORE CORRECT CAUSAL RELATIONS?

Given the effective performance of LLMs on causal discovery tasks, a pertinent research question arises: Why can LLMs perform so well? We posit that a significant factor is the nature of the pre-training data, which contains more correct causal relations than incorrect ones, leading LLMs to primarily memorize correct causal relations.

| | Precision↑ | Recall↑ | F1↑ | Accuracy↑ | Predict edges | NHD↓ | Baseline NHD | Ratio (NHD/Baseline NHD)↓ |
|---|---|---|---|---|---|---|---|---|
| OLMo-7b-Instruct (0 ICL) | 0.4166 | 1.0 | 0.5882 | 0.5625 | 12 | 0.4375 | 1.0 | 0.4375 |
| OLMo-7b-Instruct (1 ICL) | 0.4 | 0.8 | 0.5333 | 0.5625 | 10 | 0.4375 | 0.9375 | 0.4667 |
| OLMo-7b-Instruct (2 ICL) | 0.5 | 0.8 | 0.6153 | 0.6875 | 8 | 0.3125 | 0.8125 | 0.3846 |
| OLMo-7b-Instruct (3 ICL) | 0.5714 | 0.8 | 0.6667 | 0.75 | 7 | 0.3125 | 0.9375 | 0.3333 |
| OLMo-7b-Instruct (4 ICL) | 0.5 | 1.0 | 0.6667 | 0.6875 | 10 | 0.3125 | 0.9375 | 0.3333 |
| BLOOM-7b1 (3 ICL) | 0.4 | 0.4 | 0.4 | 0.625 | 5 | 0.375 | 0.625 | 0.6 |
| Llama2-7b-chat (3 ICL) | 0.4166 | 1.0 | 0.5882 | 0.5625 | 12 | 0.4375 | 1.0 | 0.4375 |
| Llama3-8b-Instruct (3 ICL) | 1.0 | 0.8 | **0.8889** | 0.9375 | 4 | 0.0625 | 0.5625 | **0.1111** |
| GPT-3.5-turbo (3 ICL) | 1.0 | 0.8 | **0.8889** | 0.9375 | 4 | 0.0625 | 0.5625 | **0.1111** |
| GPT-4o (3 ICL) | 0.8 | 0.8 | 0.8 | 0.875 | 5 | 0.125 | 0.625 | 0.2 |

Table 11: Causal discovery results for the Cancer causal graph, with 4 nodes and 5 edges. Llama3-8b-Instruct and GPT-3.5-turbo surpass all other models. The second-best performing model is GPT-4o.

| | Precision↑ | Recall↑ | F1↑ | Accuracy↑ | Predict edges | NHD↓ | Baseline NHD | Ratio (NHD/Baseline NHD)↓ |
|---|---|---|---|---|---|---|---|---|
| OLMo-7b-Instruct (0 ICL) | 0.4166 | 1.0 | 0.5882 | 0.5625 | 12 | 0.4375 | 1.0625 | 0.4117 |
| OLMo-7b-Instruct (1 ICL) | 0.4166 | 1.0 | 0.5882 | 0.5625 | 12 | 0.4375 | 1.0625 | 0.4117 |
| OLMo-7b-Instruct (2 ICL) | 0.4166 | 1.0 | 0.5882 | 0.5625 | 12 | 0.4375 | 1.0625 | 0.4117 |
| OLMo-7b-Instruct (3 ICL) | 0.5 | 1.0 | 0.6666 | 0.6875 | 10 | 0.3125 | 0.9375 | 0.3333 |
| OLMo-7b-Instruct (4 ICL) | 0.4545 | 1.0 | 0.625 | 0.625 | 11 | 0.375 | 1.0 | 0.375 |
| BLOOM-7b1 (3 ICL) | 0.4285 | 0.6 | 0.5 | 0.625 | 7 | 0.375 | 0.75 | 0.5 |
| Llama2-7b-chat (3 ICL) | 0.5556 | 1.0 | 0.7142 | 0.75 | 9 | 0.25 | 0.875 | 0.2857 |
| Llama3-8b-Instruct (3 ICL) | 1.0 | 0.8 | 0.8889 | 0.9375 | 4 | 0.0625 | 0.5625 | 0.1111 |
| GPT-3.5-turbo (3 ICL) | 1.0 | 1.0 | **1.0** | 1.0 | 5 | 0 | 0.625 | **0** |
| GPT-4o (3 ICL) | 0.8333 | 1.0 | 0.9091 | 0.9375 | 6 | 0.0625 | 0.6875 | 0.0909 |

Table 12: Causal discovery results for the Diabetes causal graph, with 4 nodes and 5 edges. GPT-3.5-turbo accurately predict the ground-truth causal graph. The second-best performing model is GPT-4o.

**Research Question 4.** *Do pre-training corpora contain more correct causal relations than incorrect ones?*

Humans fundamentally rely on causal relations to understand and generate text. Therefore, it is reasonable that pre-training corpora, which are collected from human-generated texts, are likely to inherently contain a higher proportion of correct causal relations.

*Observation* We count the total occurrence of correct and incorrect causal relations in Dolma and ROOTS corpora. The results are shown in Table 16. We use exact matching to count correct and incorrect causal relations. We observe that the occurrence of causal relations is, on average, 12 times higher than that of incorrect causal relations in Dolma and ROOTS corpora. From our observation, most incorrect causal relations do not exist in an affirmation context. They are usually in a question or negation context. For example, "Which option is correct? A. smoking causes cancer B. cancer causes smoking" or "Which means that either smoking causes cancer or cancer causes smoking."

*Discussion* In conclusion, these experimental results show that correct causal relations are more frequently represented than incorrect ones in pre-training corpora. This also explain why LLMs can identify many causal relations in causal discovery tasks.

### B.3 INFLUENCE OF MODEL SIZE ON LLMS' PERFORMANCE IN CAUSAL DISCOVERY TASKS

**Research Question 5.** *Do larger models perform better on causal discovery tasks?*

We assume that within the same architectural framework, increasing the model size (i.e., the number of parameters) leads to improved performance on causal discovery tasks. The rationale is that larger models can memorize more information from the pre-training data than their smaller models.

*Observation* We select models from the Llama2 and Llama3 series, each varying in size. These models are evaluated on causal discovery and causal direction identification tasks, with results documented in Table 17 and 18. The findings indicate that for both the Llama2 and Llama3 models, there is a positive correlation between the number of parameters and performance. However, discrepancies arise when comparing across architectures. For example, a small Llama3 model (*e.g.,* Llama3-8b-Instruct) can outperform a significantly larger Llama3 model (*e.g.,* Llama2-70b-chat). Notably, across most datasets, Llama3-70b-Instruct either matches or surpasses the performance of the currently leading closed-source LLM, GPT-4o.

| | Precision↑ | Recall↑ | F1↑ | Accuracy↑ | Predict edges (46) | NHD↓ | Baseline NHD | Ratio (NHD/Baseline NHD)↓ |
|---|---|---|---|---|---|---|---|---|
| OLMo-7b-Instruct (0 ICL) | 0.5714 | 0.8 | 0.6666 | 0.75 | 7 | 0.3125 | 0.9375 | 0.3333 |
| OLMo-7b-Instruct (1 ICL) | 0.5 | 1.0 | 0.6666 | 0.6875 | 10 | 0.3125 | 0.9375 | 0.3333 |
| OLMo-7b-Instruct (2 ICL) | 0.5555 | 1.0 | 0.7142 | 0.75 | 9 | 0.25 | 0.875 | 0.2857 |
| OLMo-7b-Instruct (3 ICL) | 0.8 | 0.8 | 0.8 | 0.875 | 5 | 0.125 | 0.625 | 0.2 |
| OLMo-7b-Instruct (4 ICL) | 0.5555 | 1.0 | 0.7142 | 0.75 | 9 | 0.25 | 0.875 | 0.2857 |
| BLOOM-7b1 (3 ICL) | 0.4444 | 0.8 | 0.5714 | 0.625 | 9 | 0.375 | 0.875 | 0.4285 |
| Llama2-7b-chat (3 ICL) | 0.8333 | 1.0 | **0.9091** | 0.9375 | 6 | 0.0625 | 0.6875 | **0.0909** |
| Llama3-8b-Instruct (3 ICL) | 0.8333 | 1.0 | **0.9091** | 0.9375 | 6 | 0.0625 | 0.6875 | **0.0909** |
| GPT-3.5-turbo (3 ICL) | 0.8333 | 1.0 | **0.9091** | 0.9375 | 6 | 0.0625 | 0.6875 | **0.0909** |
| GPT-4o (3 ICL) | 0.8333 | 1.0 | **0.9091** | 0.9375 | 6 | 0.0625 | 0.6875 | **0.0909** |

Table 13: Causal discovery results for the Obesity causal graph, with 4 nodes and 5 edges. Llama2-7b-chat, Llama3-8b-Instruct, GPT-3.5-turbo and GPT-4o outperform all other models. The second-best performing method is OLMo-7b-Instruct (3 ICL).

| | Precision↑ | Recall↑ | F1↑ | Accuracy↑ |
|---|---|---|---|---|
| OLMo-7b-Instruct (0 ICL) | 0.5482 | **0.8831** | 0.6765 | 0.5778 |
| OLMo-7b-Instruct (1 ICL) | 0.5491 | 0.8184 | 0.6573 | 0.5734 |
| OLMo-7b-Instruct (2 ICL) | 0.5771 | 0.7825 | 0.6643 | 0.6047 |
| OLMo-7b-Instruct (3 ICL) | 0.6612 | 0.8427 | 0.7410 | 0.7053 |
| OLMo-7b-Instruct (4 ICL) | 0.5294 | 0.8721 | 0.6589 | 0.5486 |
| BLOOM-7b1 (3 ICL) | 0.5027 | 0.7248 | 0.5937 | 0.5041 |
| Llama2-7b-chat (3 ICL) | 0.6197 | 0.7774 | 0.6897 | 0.6503 |
| Llama3-8b-Instruct (3 ICL) | 0.7659 | 0.6575 | 0.7076 | 0.7282 |
| GPT-3.5-turbo (3 ICL) | 0.6732 | 0.7308 | 0.7008 | 0.6891 |
| GPT-4o (3 ICL) | **0.8141** | 0.8342 | **0.8240** | **0.8224** |

Table 14: Causal direction identification results on the ConceptNet dataset, with 1900 causal relations and 1900 reverse causal relations. GPT-4o outperforms all competing methods, achieving an F1 score of 0.8240. The second-best performing method is OLMo-7b-Instruct (3 ICL), with an F1 score of 0.7410.

_Discussion_ The experiment results lead to a critical consideration of the 'bigger is better' paradigm in LLM research. Future research should thus not only focus on scaling up the size but also on refining the architecture and learning algorithms to better leverage increased model capacity.

| | Precision↑ | Recall↑ | F1↑ | Accuracy↑ |
|---|---|---|---|---|
| OLMo-7b-Instruct (0 ICL) | 0.5461 | **0.9657** | 0.6977 | 0.5815 |
| OLMo-7b-Instruct (1 ICL) | 0.5359 | 0.9606 | 0.6881 | 0.5644 |
| OLMo-7b-Instruct (2 ICL) | 0.5610 | 0.9091 | 0.6938 | 0.5988 |
| OLMo-7b-Instruct (3 ICL) | 0.6568 | 0.8771 | 0.7511 | 0.7094 |
| OLMo-7b-Instruct (4 ICL) | 0.5860 | 0.9410 | 0.7223 | 0.6382 |
| BLOOM-7b1 (3 ICL) | 0.5067 | 0.6928 | 0.5853 | 0.5092 |
| Llama2-7b-chat (3 ICL) | 0.7030 | 0.8931 | 0.7867 | 0.7582 |
| Llama3-8b-Instruct (3 ICL) | 0.8838 | 0.8296 | 0.8558 | 0.8602 |
| GPT-3.5-turbo (3 ICL) | **0.8990** | 0.8857 | 0.8923 | 0.8931 |
| GPT-4o (3 ICL) | 0.8596 | 0.9557 | **0.9051** | **0.8998** |

Table 15: Causal direction identification results on the CauseNet dataset, with 814 causal relations and 814 reverse causal relations. GPT-4o outperforms all competing methods, achieving an F1 score of 0.9051. The second-best performing method is GPT-3.5-turbo, with an F1 score of 0.8923.

| | | Correct Causal Relations | Incorrect Causal Relations |
|---|---|---|---|
| Causal Discovery (all datasets) | Dolma | 28812 | 1127 |
| | ROOTS | 814 | 118 |
| Causal Direction Identification (ConceptNet) | Dolma | 41407 | 3410 |
| | ROOTS | 1176 | 131 |
| Causal Direction Identification (CauseNet) | Dolma | 949427 | 107070 |
| | ROOTS | 24591 | 4236 |

Table 16: Occurrences of correct and incorrect causal relations in the Dolma and ROOTS corpora.

| | Precision↑ | Recall↑ | F1↑ | Accuracy↑ | Predict edges | NHD↓ | Baseline NHD | Ratio (NHD/Baseline NHD)↓ |
|---|---|---|---|---|---|---|---|---|
| **Arctic Sea Ice** | | | | | | | | |
| Llama2-7b-chat (3 ICL) | 0.4444 | 0.5217 | 0.48 | 0.6388 | 54 | 0.3611 | 0.6944 | 0.52 |
| Llama2-13b-chat (3 ICL) | 0.4478 | 0.6522 | 0.5309 | 0.6319 | 67 | 0.3681 | 0.7847 | 0.4690 |
| Llama2-70b-chat (3 ICL) | 0.3606 | 0.9565 | 0.5238 | 0.4444 | 122 | 0.5556 | 1.0 | 0.5556 |
| Llama3-8b-Instruct (3 ICL) | 1.0 | 0.1956 | 0.3272 | 0.7430 | 9 | 0.2569 | 0.3819 | 0.6727 |
| Llama3-70b-Instruct (3 ICL) | 0.5689 | 0.7174 | **0.6346** | **0.7361** | 58 | 0.2639 | 0.7222 | **0.3653** |
| GPT-3.5-turbo (3 ICL) | 0.7647 | 0.2826 | 0.4126 | 0.7431 | 17 | 0.2569 | 0.4375 | 0.5873 |
| GPT-4o (3 ICL) | 0.5178 | 0.6304 | 0.5686 | 0.6944 | 56 | 0.3055 | 0.7083 | 0.4313 |
| **Insurance** | | | | | | | | |
| Llama2-7b-chat (3 ICL) | 0.1245 | 0.7115 | 0.2120 | 0.6227 | 297 | 0.3772 | 0.4787 | 0.7879 |
| Llama2-13b-chat (3 ICL) | 0.1338 | 0.7307 | 0.2262 | 0.6433 | 284 | 0.3566 | 0.4609 | 0.7738 |
| Llama2-70b-chat (3 ICL) | 0.1619 | 0.7692 | 0.2675 | 0.6995 | 247 | 0.3004 | 0.4102 | 0.7324 |
| Llama3-8b-Instruct (3 ICL) | 0.2656 | 0.3269 | 0.2931 | 0.8875 | 64 | 0.1124 | 0.1591 | 0.7069 |
| Llama3-70b-Instruct (3 ICL) | 0.2183 | 0.5961 | 0.3195 | 0.8189 | 142 | 0.1811 | 0.2661 | 0.6804 |
| GPT-3.5-turbo (3 ICL) | 0.1575 | 0.5 | 0.2396 | 0.7736 | 165 | 0.2263 | 0.2976 | 0.7603 |
| GPT-4o (3 ICL) | 0.2287 | 0.6730 | **0.3414** | 0.8148 | 153 | 0.1851 | 0.2812 | **0.6585** |
| **Alcohol** | | | | | | | | |
| Llama2-7b-chat (3 ICL) | 0.75 | 1.0 | 0.8571 | 0.8889 | 4 | 0.1111 | 0.7778 | 0.1429 |
| Llama2-13b-chat (3 ICL) | 0.75 | 1.0 | 0.8571 | 0.8889 | 4 | 0.1111 | 0.7778 | 0.1429 |
| Llama2-70b-chat (3 ICL) | 0.75 | 1.0 | 0.8571 | 0.8889 | 4 | 0.1111 | 0.7778 | 0.1429 |
| Llama3-8b-Instruct (3 ICL) | 1.0 | 1.0 | **1.0** | 1.0 | 3 | 0 | 0.6667 | **0** |
| Llama3-70b-Instruct (3 ICL) | 1.0 | 1.0 | **1.0** | 1.0 | 3 | 0 | 0.6667 | **0** |
| GPT-3.5-turbo (3 ICL) | 1.0 | 1.0 | **1.0** | 1.0 | 3 | 0 | 0.6667 | **0** |
| GPT-4o (3 ICL) | 1.0 | 1.0 | **1.0** | 1.0 | 3 | 0 | 0.6667 | **0** |
| **Cancer** | | | | | | | | |
| Llama2-7b-chat (3 ICL) | 0.4166 | 1.0 | 0.5882 | 0.5625 | 12 | 0.4375 | 1.0 | 0.4375 |
| Llama2-13b-chat (3 ICL) | 0.5556 | 1.0 | 0.7143 | 0.75 | 9 | 0.25 | 0.875 | 0.2857 |
| Llama2-70b-chat (3 ICL) | 0.5556 | 1.0 | 0.7143 | 0.75 | 9 | 0.25 | 0.875 | 0.2857 |
| Llama3-8b-Instruct (3 ICL) | 1.0 | 0.8 | **0.8889** | 0.9375 | 4 | 0.0625 | 0.5625 | **0.1111** |
| Llama3-70b-Instruct (3 ICL) | 1.0 | 0.8 | **0.8889** | 0.9375 | 4 | 0.0625 | 0.5625 | **0.1111** |
| GPT-3.5-turbo (3 ICL) | 1.0 | 0.8 | **0.8889** | 0.9375 | 4 | 0.0625 | 0.5625 | **0.1111** |
| GPT-4o (3 ICL) | 0.8 | 0.8 | 0.8 | 0.875 | 5 | 0.125 | 0.625 | 0.2 |
| **Diabetes** | | | | | | | | |
| Llama2-7b-chat (3 ICL) | 0.5556 | 1.0 | 0.7142 | 0.75 | 9 | 0.25 | 0.875 | 0.2857 |
| Llama2-13b-chat (3 ICL) | 0.625 | 1.0 | 0.7692 | 0.8125 | 8 | 0.1875 | 0.8125 | 0.2307 |
| Llama2-70b-chat (3 ICL) | 0.625 | 1.0 | 0.7692 | 0.8125 | 8 | 0.1875 | 0.8125 | 0.2307 |
| Llama3-8b-Instruct (3 ICL) | 1.0 | 0.8 | 0.8889 | 0.9375 | 4 | 0.0625 | 0.5625 | 0.1111 |
| Llama3-70b-Instruct (3 ICL) | 1.0 | 1.0 | **1.0** | 1.0 | 5 | 0 | 0.625 | **0** |
| GPT-3.5-turbo (3 ICL) | 1.0 | 1.0 | **1.0** | 1.0 | 5 | 0 | 0.625 | **0** |
| GPT-4o (3 ICL) | 0.8333 | 1.0 | 0.9091 | 0.9375 | 6 | 0.0625 | 0.6875 | 0.0909 |
| **Obesity** | | | | | | | | |
| Llama2-7b-chat (3 ICL) | 0.8333 | 1.0 | **0.9091** | 0.9375 | 6 | 0.0625 | 0.6875 | **0.0909** |
| Llama2-13b-chat (3 ICL) | 0.8333 | 1.0 | **0.9091** | 0.9375 | 6 | 0.0625 | 0.6875 | **0.0909** |
| Llama2-70b-chat (3 ICL) | 0.8333 | 1.0 | **0.9091** | 0.9375 | 6 | 0.0625 | 0.6875 | **0.0909** |
| Llama3-8b-Instruct (3 ICL) | 0.8333 | 1.0 | **0.9091** | 0.9375 | 6 | 0.0625 | 0.6875 | **0.0909** |
| Llama3-70b-Instruct (3 ICL) | 0.8333 | 1.0 | **0.9091** | 0.9375 | 6 | 0.0625 | 0.6875 | **0.0909** |
| GPT-3.5-turbo (3 ICL) | 0.8333 | 1.0 | **0.9091** | 0.9375 | 6 | 0.0625 | 0.6875 | **0.0909** |
| GPT-4o (3 ICL) | 0.8333 | 1.0 | **0.9091** | 0.9375 | 6 | 0.0625 | 0.6875 | **0.0909** |

Table 17: Performance on causal discovery task using Llama2 and Llama3 models of different sizes.

| ConceptNet | | | |
|---|---|---|---|
| **Precision↑** | **Recall↑** | **F1↑** | **Accuracy↑** |
| Llama2-7b-chat (3 ICL)    0.6197 | 0.7774 | 0.6897 | 0.6503 |
| Llama2-13b-chat (3 ICL)   0.6010 | 0.8605 | 0.7077 | 0.6647 |
| Llama2-70b-chat (3 ICL)   0.6384 | 0.8742 | 0.7380 | 0.6897 |
| Llama3-8b-Instruct (3 ICL)   0.7659 | 0.6575 | 0.7076 | 0.7283 |
| Llama3-70b-Instruct (3 ICL)   0.8555 | 0.8253 | **0.8401** | **0.8430** |
| GPT-3.5-turbo (3 ICL)   0.6732 | 0.7308 | 0.7008 | 0.6891 |
| GPT-4o (3 ICL)   0.8141 | 0.8342 | 0.8240 | 0.8224 |
| CauseNet | | | |
| **Precision↑** | **Recall↑** | **F1↑** | **Accuracy↑** |
| Llama2-7b-chat (3 ICL)    0.7030 | 0.8931 | 0.7867 | 0.7582 |
| Llama2-13b-chat (3 ICL)   0.6625 | 0.9213 | 0.7708 | 0.7260 |
| Llama2-70b-chat (3 ICL)   0.7359 | 0.9521 | 0.8302 | 0.8053 |
| Llama3-8b-Instruct (3 ICL)   0.8838 | 0.8296 | 0.8558 | 0.8602 |
| Llama3-70b-Instruct (3 ICL)   0.8939 | 0.9423 | **0.9175** | **0.9152** |
| GPT-3.5-turbo (3 ICL)   0.8990 | 0.8857 | 0.8923 | 0.8931 |
| GPT-4o (3 ICL)   0.8596 | 0.9557 | 0.9051 | 0.8998 |

Table 18: Performance on causal direction identification task using Llama2 and Llama3 models of different sizes.