# OpenReview forum: "From Corpora to Causality: Unveiling Causal Comprehension in Large Language Models"
_ICLR.cc/2025/Conference — ICLR 2025 Conference Withdrawn Submission_

### Official Review · Reviewer_LYZv · 2024-10-26

**Soundness:** 3
**Presentation:** 2
**Contribution:** 2
**Rating:** 5
**Confidence:** 4

**Summary:**

This paper presents an interesting study and aims to understand the ability of LLMs for causal discovery. Particularly, the authors utilised open-sourced (open weight and open data) LLMs for better experimental control, thus provide more convincing conclusion compared to studies based on close-source models. The experimental results suggest that LLMs are primarily conducting causal inference tasks based on memorisation from pre-training data, rather than doing "generalisation and reasoning". They also suggest that the effect of causal discovery performance are strongly associated with the context, shed lights on how we should train, adopt and deploy LLMs for causal discovery in practice.

**Strengths:**

The paper presents an very interesting study and made an important claim that they used open-sourced models compared to previous study with closed source models. Here are the strengths listed below:

1. experimental methods (open source v.s. close source): this is the main novelty of this paper since open source models allow us to make more convincing claims
2. using a mixture of real-world (concept net and causal net) and synthetic dataset: which makes the conclusion more promising
3. have a good amount of experiments with relevant analysis: the authors provide enough experiments on three research questions with somewhat detailed analysis of the results

**Weaknesses:**

This paper have the following weakness:

1. experimental design: the authors suggest that it is hard to construct causal relationships from pre-training data, however, this is one of the main motivation for us to adopt experiment with an open-source model, right? for the real-world data, I think the authors could build two sets of data (as similar to A2) by first construct the top confidence causal graph and then make two variations (one is mentioned in pre-trained data and one is not) based on cross-reference via some sort of similarity research with the pre-trained data used for OLMo. I agree the second synthetic experiments are similar to this idea, however I am not sure fine-tuning with LoRA will gives a similar level of ability as the original OLMo model.

2.presentation and writing style: the general presentation and writing style sound like a report or a coursework and does not seems to be very scientific. Especially, I think whenever claims on "xxxx ability" is made, the authors are advised to give a solid definition or a way to quantify these claims. In addition, the discussion should contains more details and explanation of the results.

**Questions:**

Q1: please refers to the experimental design part of the weakness section above, I would like to know what the authors opinions?

Q2: I am not sure if the paper is over-claiming things, the authors seems to be converting causal discovery task as classification tasks (1. if A causes B and 2. full causal relationship in a CDAG). This is not the same as doing causal discovery since type 1 is a simplified version between only two variables and type 2 should typically involves iteratively refinements of the DAG structure (e.g. the classical PC algorithm as an example). I would suggest the authors to narrow down the scope of their claim to better reflect on the actual contribution.

---

### Official Review · Reviewer_5oHR · 2024-11-03

**Soundness:** 2
**Presentation:** 3
**Contribution:** 2
**Rating:** 3
**Confidence:** 4

**Summary:**

This paper investigates the effectiveness of Large Language Models (LLMs) in causal discovery, using open-source models OLMo-7b-Instruct and BLOOM-7b1. The authors explore three key research questions (RQ):
1. The extent to which LLMs rely on memorization vs. generalization,
2. The impact of incorrect causal relations in pre-training data, and
3. How contextual information (causal information, e.g., rain causes flooding.) influences causal inference.

Their study finds that LLMs are proficient in identifying frequently occurring causal relations but struggle with generalizing rare or novel relations, emphasizing a dependency on memorization (Addresses RQ 1). Incorrect causal information in pre-training data notably reduces LLM confidence in correct causal relations (Addresses RQ 2), and context plays a critical role in LLM performance, suggesting a need for careful management of training data and contextual cues (Addresses RQ 3).

**Strengths:**

1. Good RQs – The paper addresses significant aspects of LLM-based causal discovery from data, namely memorization of the data, the effect of incorrect causal relationships in the data, and the role of contextual cues, which together provide a good overall evaluation of LLM capabilities, within the scope of this work.

2. Informative Findings – The results are intuitive and practical, suggesting that high occurrences of correct causal data enhance LLM performance, while conflicting or negative contexts degrade accuracy. This provides good guidance for improving LLM training.

3. Use of Synthetic Data – Including synthetic data to assess LLM generalization without memorization biases is innovative and interesting.

**Weaknesses:**

Part A: Significant theory-related weaknesses:

1. What precisely is memorization vs. generalizability in causal discovery? The paper empirically assesses memorization and generalization but lacks a formal theoretical framework that quantifies or distinguishes the two modes of inference. Specifically, while correlations between occurrence frequencies and F1 scores offer insight, they do not necessarily capture the structure of causal relations. This omission limits the analysis to surface-level observations without a rigorous theoretical basis for why certain relations are memorized while others require generalization.

2. Need to assess noise-related effects before assessing the negative impact of incorrect causal relations or positive impact of good contextual cues – Both the analysis of correct and incorrect causal relations lacks theoretical models that can help estimate the effect of improvements, detriments, or independent noise in causal inference. For instance, while increased occurrences of reverse causal relations reduce LLM confidence, the study does not quantify how varying noise levels impact predictive accuracy or stability. Without a mathematical model of error propagation, the conclusions remain qualitative.

Part B: Significant practical weakness:

Limited Model Scope: Although the study uses open-source models, it lacks comparison with more advanced closed-source LLMs like GPT-4, which may have different generalization properties. This restricts the applicability of findings to open-source models only. If the theory from the previous sections were handled properly, this step may have been less important.

Summary of Weaknesses:
Despite valuable findings, the paper's lack of theoretical depth and limited model scope prevent it from making a substantial contribution at this stage. Strengthening the theoretical foundations and addressing these limitations in future work could make this study a valuable contribution to LLM-based causal discovery research.

**Questions:**

It would be great if the authors could provide additional insight on the questions below, even if it is only empirically substantiated to a limited extent, for the moment:

1. Model Generalization: Could hybrid models that integrate statistical causal discovery with LLMs improve generalization for causal discovery? How would such integration impact performance on low-frequency relations?

2. Effectiveness of Synthetic Data: How does the model's performance differ when synthetic data is used versus real-world data? Could the findings on memorization versus generalization change with a larger synthetic dataset?

3. Impact of Conflicting Information: Given that incorrect causal relations reduce confidence, are there data augmentation techniques that could help LLMs better handle conflicting information?

4. Contextual Variability: How would the performance of LLMs change if contextual cues were randomized or misaligned in training versus inference stages? Would this variability affect the model's robustness in real-world scenarios where context can be dynamic?

---

### Official Review · Reviewer_ps2t · 2024-11-03

**Soundness:** 1
**Presentation:** 2
**Contribution:** 1
**Rating:** 3
**Confidence:** 3

**Summary:**

This paper studies memorization of causal relations in large language models; the authors constructed a synthetic dataset with causal annotations from Dolma, a pre-training corpse, and performed studies on several large language models.

**Strengths:**

The problem of the efficacy of causal reasoning of LLMs itself is intereting and the authors' idea of studying this through memorization is also novel.

**Weaknesses:**

My main concerns is that this paper looks rush and the authors could have dive deeper; furthermore I have a concern on the methodology the authors constructed the dataset.

1. The authors restricted their attention to causal direction identification only, which I think is fine. But it's unclear to me how the authors could identify the bias due to correlations. For example, it is not hard to construct examples of $(X,Y)$ where both direction makes sense and might found in the corpus. For example, let X be "John studies hard" and Y be "John is under pressure;" John could be under pressure because he studies hard or he could study hard whenever he's under pressure.

2. I do not think the second causal query through textual order make sense. Specifically, the order events are written in the text, the time events occur, and the direction of causal effect are in general all different. I am suspecting this may introduce systematic bias in the authors' methodology.

3. It's also unclear what the results in Table 1 on p 9 suggest -- for one thing I do not see any quantification of error such as stderr; in additional, could the authors perhaps provide more dicussions on "how large the affirmative ratio is good enough?"

Minor stylish issues:

1. Line 047, 104. double quotes surrounding "causal parrots." Similar issue at line 243.

2. Captions and axes tickers of all plots are too small.

**Questions:**

See Weaknesses.

---

### Official Review · Reviewer_DAez · 2024-11-04

**Soundness:** 2
**Presentation:** 3
**Contribution:** 2
**Rating:** 5
**Confidence:** 3

**Summary:**

This paper investigates LLM's ability in performing causal discovery. It studies whether LLM's ability for identifying causal relations only come from memorization of training data, and to what extent it can be generated to identify new relations. It also studies how LLM's causal discovery ability will be affected by incorrect causal relationships in training data as well as the context of causal relations. The evaluation of LLM on these questions are performed through both synthetic causal relations or causal relations from real data.

**Strengths:**

- the questions studied in this paper are very relevant and important to causal related applications of LLM

- this paper is one of the first to use training data of LLM to studying the memorization effect on causal

- the presentation of this paper makes it easy to read

**Weaknesses:**

- When generating the synthetic data in Section 4.1.2, it seems that all the causal relations are generated in a pair-wise manner, rather than generated from a graph structure. That is, the edge density of the entire causal graph, the edge degree of nodes in the graph, the number of Markov equivalent class of resulted graph are not described. Therefore, it seems the scope of synthetic data is not that relevant for causal discovery

- Furthermore, when generating synthetic data in Section 4.1.2, only synthetic relations are generated, the data part used for performing causal discovery (generated through functional causal models) is never generated. Whether relations are correct or incorrect seem to be determined by sentences of "blaonge causes goloneke" instead of the data variable distribution reflecting the causal relations. Note that in casual discovery framework, the correspondence of data and graph are crucial. That's why there are causal Markov assumptions and causal faithfulness assumptions in causal discovery framework. However,  in synthetic data of this paper, only graph edge relations are generated, and they are generated in a pair-wise manner not following any graphicial model (such as ER graph), and no data variable columns are generated. This makes this paper deviating from the scope of causal discovery, but rather a on scope of "causal sentence validating".



- Causal discovery algorithms such as PC algorithm reviewed in Section 2 is not used to compare with LLM in various experiment settings

- In literature review part of Section 2, the class of functional causal model based methods such as LiNGAM are not mentioned. Also, recent related work such as "Understanding causality with large language models: Feasibility and opportunities", "Is Knowledge All Large Language Models Needed for Causal Reasoning?", "LLM4Causal: Democratized Causal Tools for Everyone via Large Language Model", "Answering Causal Questions with Augmented LLMs" are not discussed.

**Questions:**

See all points in weakness.

---

### Official Review · Reviewer_xKzj · 2024-11-05

**Soundness:** 4
**Presentation:** 4
**Contribution:** 3
**Rating:** 8
**Confidence:** 5

**Summary:**

The paper examines how LLMs perform in causal discovery, focusing on their tendencies toward memorization, the impact of incorrect causal information, and the influence of context. Using the open-source models OLMo and BLOOM, the authors analyze how well LLMs recognize and generalize causal relations, especially for rare or novel cases. They find that LLMs rely heavily on memorized causal data are sensitive to  incorrect causal information in the training data. They also point out the importance of "positive and negative context" on accuracy.
Contributions:
* Insight into LLM memorization vs. generalization in causal discovery
* Demonstraiting the negative impact of incorrect data in training
* Analysis of the role of context

**Strengths:**

Analysis on models in the <10 billion parameter ranged that can be fine-tuned and have open training data is quite welcome and builds well on prior work that focused more on large, closed models.

"Our findings indicate that while LLMs are effective in recognizing causal relations that occur frequently in pre-training data, their ability to generalize to new or rare causal relations is limited." - This is a negative result, but a negative result in a field where the abilities of these models tends to be overstated, and thus it is a useful result.

**Weaknesses:**

I find the discussion of contexts to be somewhat unsatisfying. In the given example, causal relationship "rain causes flooding"  being attenuated by "but not in cities that have good drainage" an effect ((is-flooding)) with two interacting causal parents ((is-raining)) and ((is-flooding)). Causal graphical models by default treat sets that include nodes and their direct causes the basic unit (e.g., causal Markov property). In contrast, this analysis (and ones before it) treat pairwise relationships as the basic unit -- this is done not because it is ideal, but because it allows for evaluation of causal discovery technique with binary classification and retrieval statistics. But just because this type of evaluation suffers when their are causal interactions ("negative contexts"), it doesn't mean that the LLM is failing to learn the nuances of these interactions.

**Questions:**

Would it be possible to replicate your results on Olmo's 65B model and BLOOM's 176B model? I suspect that some would believe that larger models might learn representations that can defeat incorrect poisoning of the dataset and that can capture nuances due to context better than smaller models. Getting some results on these larger open models would address those concerns.  In other words, show that these results are not dependent on scale.

---

### Official Review · Reviewer_sBbY · 2024-11-07

**Soundness:** 2
**Presentation:** 2
**Contribution:** 2
**Rating:** 3
**Confidence:** 4

**Summary:**

This study aims to investigate the effectiveness of large language models (LLMs) in causal discovery tasks by exploring three specific research questions that address the generalization capabilities of LLMs, as well as their susceptibility to incorrect causal data and contextual information. To address these questions, the study constructs targeted datasets and experimental setups.

**Strengths:**

S1. The experimental setups in this work are well-designed, utilizing existing open-source LLMs effectively. However, the data construction process can be optimized.

S2. The experimental results also support the authors' findings.

**Weaknesses:**

W1. The questions raised in this work, when considered alongside the characteristics of supervised training, lead to predictable conclusions, offering limited novel insights or valuable findings.

W2. The research questions posed in this work center around the “causal parrots” hypothesis but fail to discuss related work on causal discovery with LLMs. For instance, studies like Gao et al. (2023) in "Is ChatGPT a Good Causal Reasoner? A Comprehensive Evaluation" identify issues such as causal hallucinations and implicit causal relations, which are not considered in this paper.

W3.  The data construction process could benefit from greater utilization of existing causal discovery datasets.

**Questions:**

1. Text data is inherently complex, and the conclusions drawn in Research Question 1 require further validation. For example, in Table 3 and Table 4 in Appendix A.4 (line 318), the authors retrieve explicit causal relationships. However, in the event causality identification (ECI) task, there are also implicit causal relationships (see Section 5.6 in Gao et al.). It is unclear whether the conclusions for Research Question 1 would hold for implicit causal relationships as well.

2. For Research Question 2, the authors construct examples of incorrect causal relationships. However, it would be more convincing to validate these findings using publicly available datasets. For instance, the EventStoryLine v0.9 (ESC) dataset contains both causal and non-causal event pairs and is referenced in Gao et al. An example from this dataset includes:

- Text: "A preliminary hearing for John Jenkin, 23, charged with the murders of his mum Alice McMeekin, 58, and sister Katie Jenkin, 20, took place at Preston Crown Court this morning."

- Causal event pair: (hearing, charged)

- Non-causal event pairs: (hearing, murders), (took place, charged)

3. For Research Question 3, the authors could consider leveraging publicly available datasets with contextual information, such as the mentioned dataset ESC.

---

### Note · Authors · 2024-12-15

I have read and agree with the venue's withdrawal policy on behalf of myself and my co-authors.